# Integrin β1/FAK/SRC signal pathway is involved in autism spectrum disorder in Tspan7 knockout rats

Shuo Pang[1,2], Zhuohui Luo[1,2], Wei Dong[1], Shan Gao[1], Wei Chen[3], Ning Liu[3], Xu Zhang[3], Xiang Gao[3], Jing Li[3], Kai Gao[3], Xudong Shi[3], Feifei Guan[3], Li Zhang[3], Lianfeng Zhang[1,2]

TSPAN7 is related to various neurological disorders including autism spectrum disorder (ASD). However, the underlying synaptic mechanism of TSPAN7 in ASD is still unclear. Here, we showed that *Tspan7* knockout rats exhibited ASD-like and ID-like behavioral phenotypes, brain structure alterations including decreased hippocampal and cortical volume, and related pathological changes including reduced hippocampal neurons number, neuronal complexity, dendritic spines, and synapse-associated proteins. Then, we found that TSPAN7 deletion interrupted the integrin β1/FAK/SRC signal pathway that was followed by the down-regulation of PSD95, SYN, and GluR1/2, which are key synaptic integrity-related proteins. Furthermore, reactivation of SRC restored the expression of synaptic integrity-related proteins in primary neurons of TSPAN7 knockout brains. Taken together, our results suggested that TSPAN7 knockout caused ASD-like and ID-like behaviors in rats and impaired neuronal synapses possibly through the down-regulation of the integrin β1/FAK/SRC signal pathway, which might be a new mechanism on regulation of synaptic proteins expression and on ASD pathogenesis by mutated TSPAN7. These findings provide novel insights into the role of TSPAN7 in psychiatric diseases and highlight integrin β1/FAK/SRC as a potential target for ASD therapy.

## Introduction

Autism spectrum disorder (ASD) is a complex, highly genetically heterogeneous neurodevelopmental disorder with comorbidities, frequently associated with intellectual disability (ID) comorbidities, affecting 1–2% of children worldwide (Lord et al, 2018; Persson et al, 2020; Qin et al, 2021).

ASD is diagnosed before the age of 3 yr and remains throughout life (Berkel et al, 2012). The main specific autistic characteristics are social communication deficits, interest deficits, repetitive stereotyped behavior, and mental retardation in some cases (Sharma et al, 2018). Even though a specific etiology can be identified in some individuals with ASD, most of the variation that increase likelihood of ASD is believed to have its origin in a complex interaction between genetic and environmental factors (Bai et al, 2019; Bottema-Beutel et al, 2020; Hegarty et al, 2020). Genetic factors included copy number variations, single-nucleotide polymorphisms, and epigenetic alterations of genes such as E3 ubiquitin ligase-A (*Ube3a*), neuroligin-3 (*Nlgn3*), SH3 and multiple ankyrin repeat domains 3 (*Shank3*), methyl-CpG–binding protein 2 (*Mecp2*), phosphatase and tensin homolog deleted on chromosome 10 (*Pten*), tuberous sclerosis complex subunit 1 (*Tsc1*), and tuberous sclerosis complex subunit 2 (*Tsc2*) (Jamain et al, 2003; Varghese et al, 2017; Xu et al, 2018; Lee et al, 2021). A variety of environmental factors had been found to increase the likelihood of ASD, including premature birth, fetal exposure to psychotropic drugs, or pesticides (Sharma et al, 2018). An X-linked gene *Tspan7* (also named *Tm4sf2*) had been found to be associated to developing ASD, which encodes for tetraspanin7 (TSPAN7) protein (Noor et al, 2009; Piton et al, 2011).

TSPAN7 is a highly expressed quadruple transmembrane glycoprotein in the brain (Hemler, 2005). Under physiological conditions, TSPAN7 regulated the transport of alpha-amino-3-hydroxy-5-methyl-4-isoxazolepropionic acid (AMPA) receptors, which was involved in synaptic transmission, neuronal morphogenesis, DCs, and osteoblast morphogenesis (Bassani et al, 2012; Perot & Ménager, 2020). TSPAN7 was associated with the development of a variety of diseases, including psychiatric disorders and type 1 diabetes in pathological conditions (Zemni et al, 2000; Abidi et al, 2002; Noor et al, 2009; McLaughlin et al, 2016; Walther et al, 2016).

Previous studies found that TSPAN7 was involved in spine maturation in cultured hippocampal neurons through a direct interaction with PICK1 (Bassani et al, 2012). *Tspan7* knockout caused an ASD-related phenotype in mice, and the deletion of TSPAN7 altered voltage-gated ion channels function and decreased PKC-ERK signaling (Murru et al, 2021). Synaptic dysfunction had been revealed to be a hallmark of ASD (Guang et al, 2018). However,

[1]Key Laboratory of Human Disease Comparative Medicine, National Health Commission of China, Institute of Laboratory Animal Science, Peking Union Medical College, Chinese Academy of Medical Sciences, Beijing, China    [2]Neuroscience Center, Chinese Academy of Medical Sciences, Beijing, China    [3]Beijing Engineering Research Center for Experimental Animal Models of Human Diseases, Institute of Laboratory Animal Science, Peking Union Medical College, Chinese Academy of Medical Sciences, Beijing, China

Correspondence: zhangl@cnilas.org; zhanglf@cnilas.org

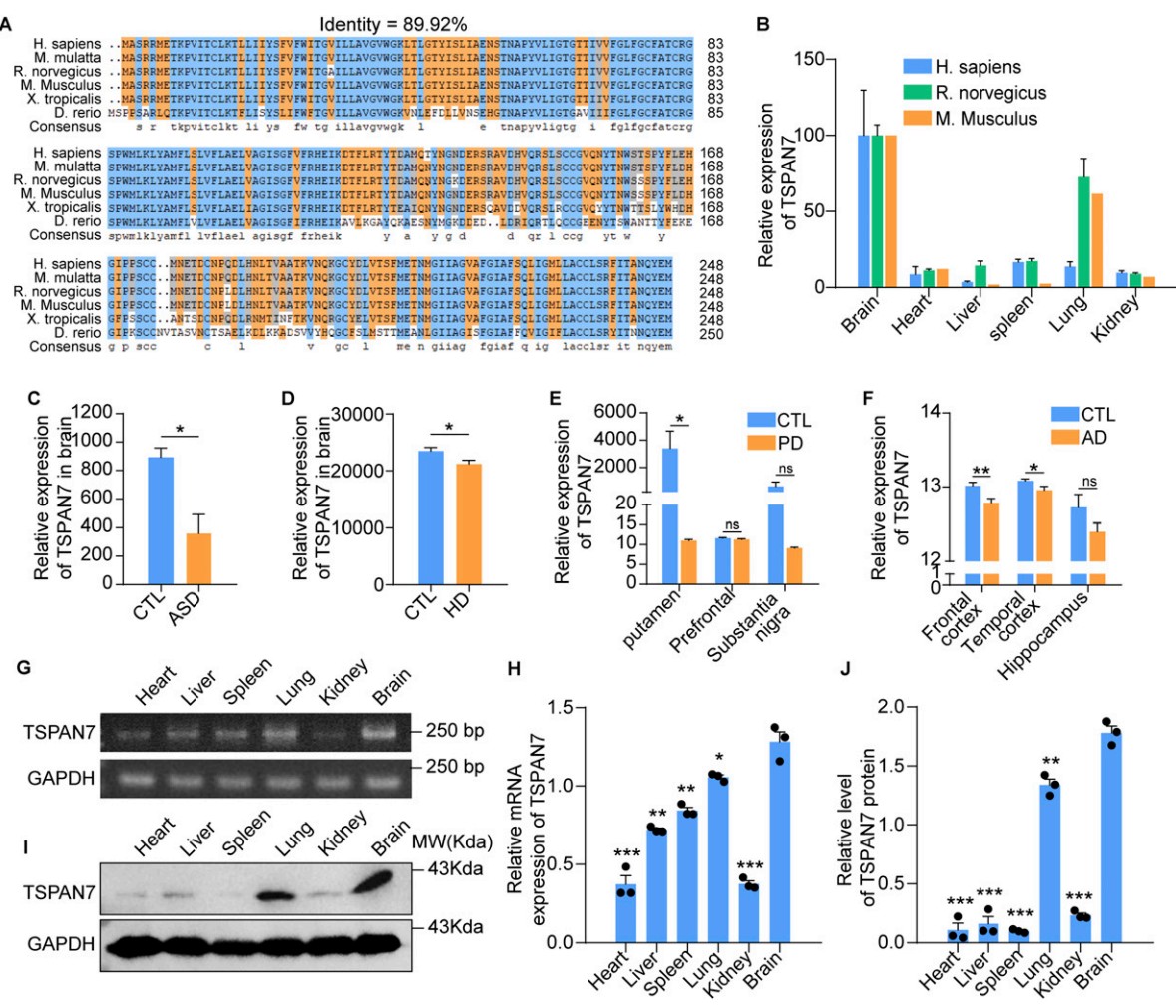

**Figure 1.  Bioinformatics analysis of TSPAN7.**
**(A)** TSPAN7 protein sequences in different species were obtained from the GenBank database, and the identity of TSPAN7 proteins in various species of *H. sapiens*, *M. mulatta*, *R. norvegicus*, *M. musculus*, *X. tropicalis*, and *D. rerio* were analyzed using the alignment tool of DNAMAN and ClustalX. **(B)** mRNA expression of TSPAN7 in different tissues were compared among the three species of human, rats, and mice, where the data were obtained from the GenBank database and normalized by the data of brain tissues. **(C, D, E, F)** Differential expression of TSPAN7 in the brains of ASD, HD, AD, and PD patients were compared with control (CTL), where data were obtained from the Gene Expression Omnibus database. **(G, H, I, J)** mRNA level of *Tspan7* and protein expression of TSPAN7 in the heart, liver, spleen, lung, kidney, and brain tissues from WT Wistar rats at age of 2 mo were detected by RT–PCR and Western blot and quantified by ImageJ. n = 3, *P < 0.05, **P < 0.01, ***P < 0.001 versus brain. Data information: In (C, D, E, F), data were analyzed by two-tailed unpaired *t* tests. Error bars represent SEM. *P < 0.05, **P < 0.01, n.s, not significant.
Source data are available for this figure.

the underlying mechanisms of TSPAN7 in ASD at the synapse and synaptic integrity level remain unclear. To investigate the effect of TSPAN7 deficiency on brain development and its possible mechanisms, we generated a *Tspan7* knockout rat line using CRISPR/Cas9 technology.

The *Tspan7*⁻/⁻ rats displayed ASD-like phenotypes, such as increased repetitive behaviors, anhedonic-like state, altered sociability, and impaired learning and memory similar to mice. Interestingly, *Tspan7*⁻/⁻ rats showed obvious brain structure alteration and pathological changes, such as decrease in hippocampal and cortical volume, loss of hippocampal pyramidal neurons, and the reduction in synaptic density, which was not reported in *Tspan7*⁻/y mice. More importantly, we found that deletion of TSPAN7 protein interrupted the distribution of integrin *β*1 on the neuron membrane;

subsequently reduced integrin *β*1/FAK/SRC signaling; and downregulated the expression of synaptic associated proteins, such as postsynaptic density 95 (PSD95), synaptophysin (SYN), and glutamate reportor 1/2 (GluR1/2). These founding provided a new clue to understand the function and the importance of TSPAN7 in the development of the brain and cerebral pathogenesis of ASD.

# Results

## Bioinformatics analysis of TSPAN7

The TSPAN7 amino acid sequences among human, macaque, rat, mouse, clawed frog, and zebrafish were compared by the alignment

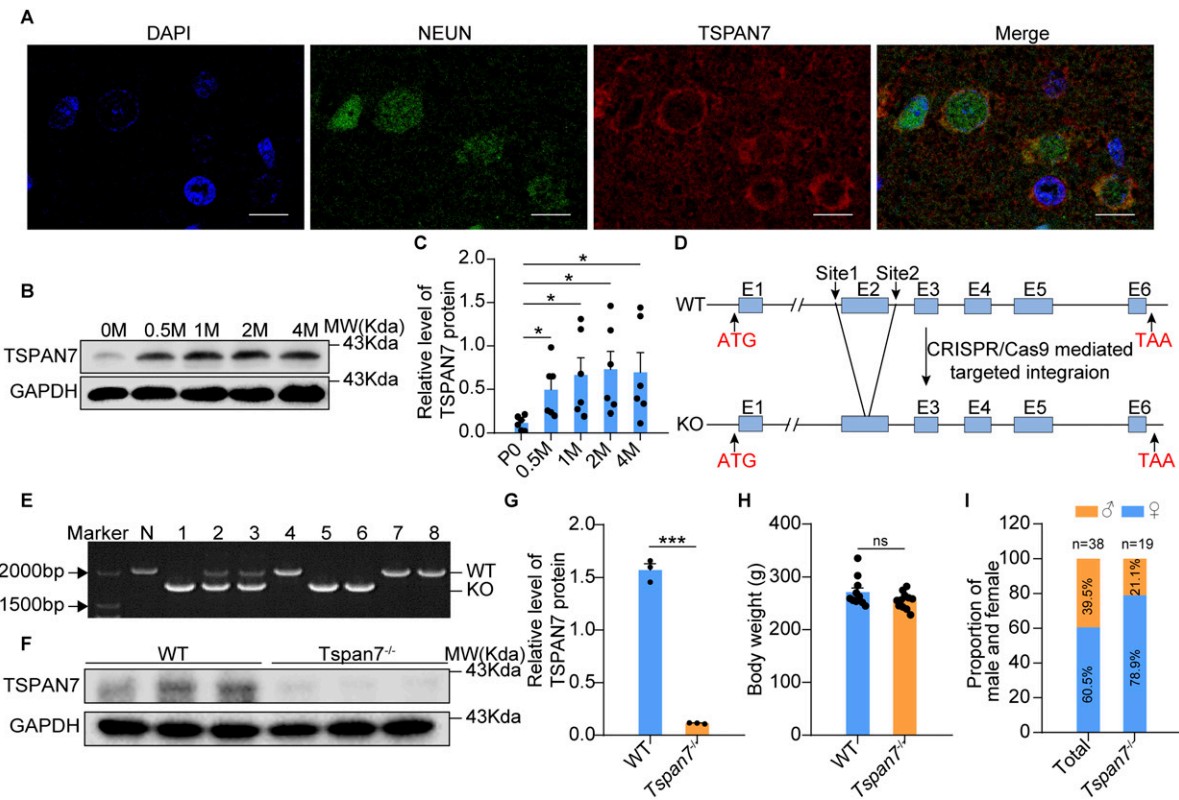

**Figure 2. Determination of the *Tspan7* expression pattern in the rat brain and the generation of *Tspan7*$^{-/-}$ rats.**
**(A)** Paraffin sections of the brain from WT Wistar rats at 6 mo old were stained by double immunofluorescence with antibodies of anti-NEUN (green) and anti-TSPAN7 (red). The images showed that TSPAN7 was expressed in mature neurons of the rat brain (Scale bar = 10 $\mu$m). **(B, C)** Protein expression level of TSPAN7 in the whole-brain tissue from WT rats at age of 0 d, 0.5, 1, 2, and 4 mo was detected by Western blot with the anti-TSPAN7 and quantified with ImageJ. n = 6, *$P$ < 0.05. **(D)** Exon 2 of *Tspan7* was knocked out as shown in the strategy of the CRISPR/Cas9 method. **(E)** Positive pups were genotyped by tail DNA PCR. N was negative, indicating WT. Numbers 4, 7, and 8 are WT; numbers 2 and 3 are heterozygotes; numbers 1, 5, and 6 are homozygotes (n = 8). **(F, G)** Deletion of TSPAN7 protein in brain tissues of *Tspan7*$^{-/-}$ rats was detected by Western blot with the anti-TSPAN7 and quantified with ImageJ (n = 3). **(H, I)** Body weights and birth rates were compared between WT and *Tspan7*$^{-/-}$ rats at 6 mo old. Data information: In (G), data were analyzed by two-tailed unpaired *t* tests. Error bars represent SEM. ***$P$ < 0.001, n.s, not significant.
Source data are available for this figure.

tool in the database of BlastP, GenBank, PubMed, and Ensembl. Strikingly, 89.92% identity among the human and five kinds of chordates was delineated (Fig 1A) and suggested that *Tspan7* was an evolutionarily conserved gene. The expression data from database of GenBank were normalized by brain tissue and showed a similar expression pattern in different tissues among the three species of human, rats, and mice. *Tspan7* was highly expressed in tissues of the brain and lung and lowly in tissues of the spleen, heart, kidney, and liver (Fig 1B). The published datasets also showed that the expression of TSPAN7 was decreased in the brain tissues from patients of ASD, Huntington's disease (HD), Parkinson's disease (PD), and Alzheimer's disease (AD) (Fig 1C–F, *$P$ < 0.05, **$P$ < 0.01, n.s, not significant). Finally, the TSPAN7 expression in different tissues was confirmed by RT–PCR and Western blot, and the results showed that TSPAN7 expression was the highest in the rat brain (Fig 1G–J, n = 3, *$P$ < 0.05, **$P$ < 0.01 and ***$P$ < 0.001 versus brain).

## Generation and general observation of *Tspan7*$^{-/-}$ rats

Immunofluorescence analysis showed that TSPAN7 was expressed in NEUN-stained neurons of the rat brain (Fig 2A). The Western blot

indicated high-level expression of TSPAN7 proteins in the rat brain from 0.5 to 4 mo of age and the relative lower expression of TSPAN7 at birth (Fig 2B and C, n = 6, *$P$ < 0.01). We used the CRISPR/Cas9 technique to knock out the *Tspan7* ortholog in rats and acquired two *Tspan7*$^{-/y}$ founders. One *Tspan7*$^{-/y}$ founder was selected to breed and obtain *Tspan7* knockout rats, which had a deletion of a 546-bp fragment containing exon 2 and part of intron 2 of the *Tspan7* ortholog (Fig 2D). The deletion caused TSPAN7 protein ablation, and Western blot analysis confirmed the absence of TSPAN7 protein in the brains of *Tspan7* KO rats (Fig 2E–G, n = 3). No obvious changes of body weight between WT and *Tspan7* KO rats were detected (Figs 2H and S1G, n.s). However, the ratio between male (*Tspan7*$^{-/y}$) to female (*Tspan7*$^{-/-}$) rats was reduced significantly (Fig 2I).

## *Tspan7*$^{-/-}$ rats displayed ASD-like behaviors

Because of the low birth rate of *Tspan7*$^{-/y}$ male rats, we used *Tspan7*$^{-/-}$ rats to perform the experiments in this current study. The five behavior tests were performed here. First, the *Tspan7*$^{-/-}$ rats displayed abnormal self-grooming behavior indicated with

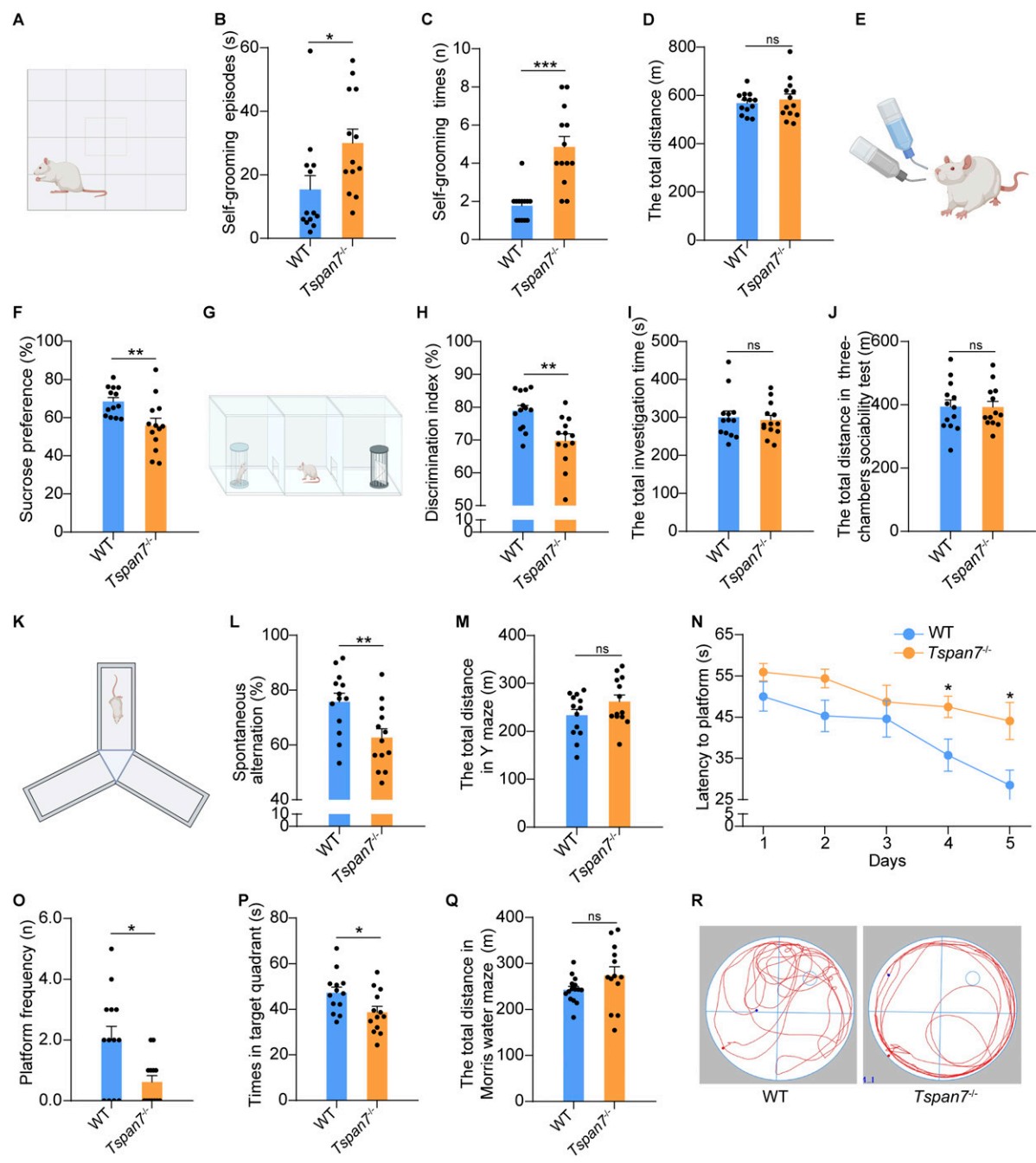

**Figure 3.  Behavior analysis.**
**(A, B, C, D)** Female WT and *Tspan7*⁻/⁻ rats at 6 mo old were first applied to the self-grooming test. The self-grooming episodes, self-grooming times, and the total distance were recorded. **(E, F)** Rats were then applied to the sucrose preference test, and uptake percentage of 1% sucrose solution was calculated and compared between WT and *Tspan7*⁻/⁻ rats. **(G, H, I, J)** Three-chamber sociability test was performed. The discrimination index for sociability (stranger versus object cage), the total investigation time, and distance were recorded and analyzed. **(K, L, M)** Y-maze test was performed. The spontaneous alternation (%) and total distance were, respectively, calculated and compared between WT and *Tspan7*⁻/⁻ rats. **(N, O, P, Q, R)** Morris water maze test was performed, and the latency to find the hidden platform during the five training days, the platform frequency, duration in target quadrant, the total movement, and swimming track on day 6 were compared between WT and *Tspan7*⁻/⁻ rats. Data information: In (B, C, E, G, I, K, L, M), data were analyzed by two-tailed unpaired *t* tests. Error bars represent SEM. n = 13, *P < 0.05, **P < 0.01, ***P < 0.001, ns, not significant. Diagrams of (A, D, F, H, J) were created with BioRender.com.
Source data are available for this figure.

increase of self-grooming episodes and times and no significant change in the total distance (Fig 3A–D, n = 13, *P < 0.05, ***P < 0.001, ns). Second, *Tspan7*⁻/⁻ rats reduced by 6.7% uptake of

sucrose water (1%) compared with the WT rats, which naturally displayed a preference for respect 1% sucrose to the sole water (Fig 3E and F, n = 13, **P < 0.01). Third, *Tspan7*⁻/⁻ rats displayed

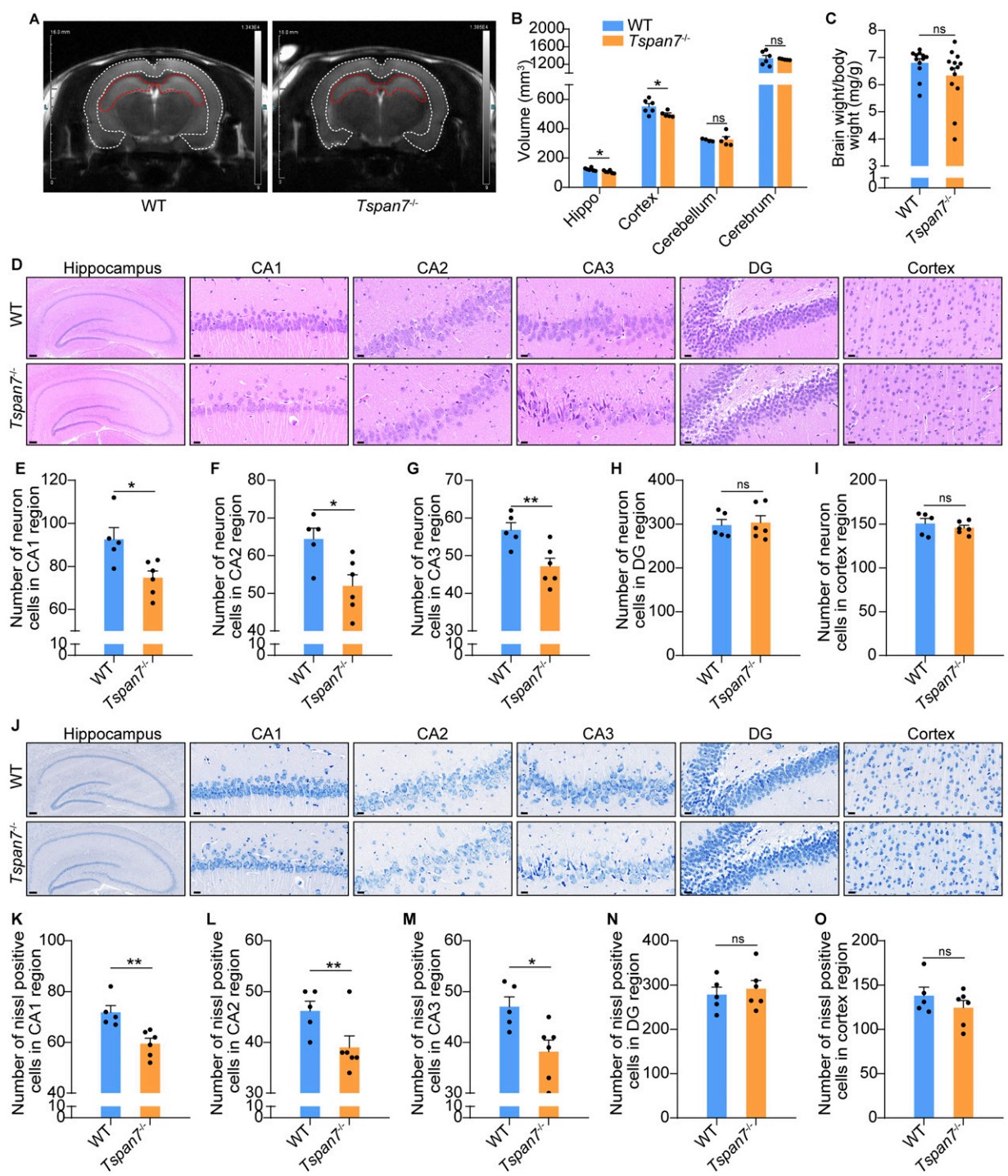

**Figure 4. Analysis of the pathological changes in the brain of *Tspan7*⁻/⁻ rats.**
(A, B) Female WT and *Tspan7*⁻/⁻ rats at 6 mo old were anesthetized and examined on an MRI system, and the volume of the hippocampus, cortex, cerebellum, and cerebrum was analyzed on the working station; n = 6 in the WT group and n = 5 in the *Tspan7*⁻/⁻ group. (C) Rats were euthanized, and the whole brains were stripped out, and the ratios of brain/body weight were calculated; n = 12 each group. (D) Coronal paraffin sections of brains were stained with the H&E method, and the images of the hippocampus and cortex were captured; scale bar = 20 $\mu$m. (E, F, G, H, I) Neurons were counted in the hippocampus of CA1, CA2, CA3, and DG regions and the cortex; n = 5 in the WT group and n = 6 in the *Tspan7*⁻/⁻ group. (J) Sections of brains were also observed by Nissl staining; scale bar = 20 $\mu$m. (K, L, M, N, O) Nissl-positive cells (blue) were counted in the hippocampus of CA1, CA2, CA3, and DG regions and the cortex; n = 5 in the WT group and n = 6 in the *Tspan7*⁻/⁻ group. Data information: In (B, C, F, G, H, I, J), data were analyzed by two-tailed unpaired $t$ tests. Error bars represent SEM. *$P < 0.05$, **$P < 0.01$, n.s, not significant.
Source data are available for this figure.

inactive social behavior indicated with spending less time investigating the stranger rats compared with that of WT rats in the three-chamber sociability test (Fig 3G and H, n = 13, **$P$ < 0.01). However, there was no significant difference in the total investigation time and distance between the $Tspan7^{-/-}$ rats and WT rats (Fig 3I and J, n = 13, ns). Moreover, $Tspan7^{-/-}$ rats showed impaired working memory indicated with the decrease of spontaneous alternations in the Y-maze test (Fig 3K and L, n = 13, **$P$ < 0.01) and did not change the total distance (Fig 3M, n = 13, ns). Finally, in the Morris water maze test, $Tspan7^{-/-}$ rats displayed an increased latency to find the hidden platform during the acquisition phase (Fig 3N, n = 13, *$P$ < 0.05). During the probe test, $Tspan7^{-/-}$ rats displayed abnormal spatial memory indicated with a significant reduction of the platform crossing number and target quadrant duration compared with WT rats (Fig 3O, P, and R, n = 13, *$P$ < 0.05). However, there is no significant difference in the total movement between the $Tspan7^{-/-}$ rats and WT rats (Fig 3Q, n = 13, ns).

### TSPAN7 deficiency caused brain structure changes

The geometry changes of the brain in $Tspan7^{-/-}$ rats were first examined by magnetic resonance imaging (MRI) and showed a significant decrease in hippocampal and cortical volume in $Tspan7^{-/-}$ brains compared with those of WT rats, whereas no obvious difference in cerebral and cerebellar volumes was detected (Fig 4A and B, n = 6, *$P$ < 0.05, ns). The gross anatomy examination showed no significant change in the brain coefficient (brain weight/body weight) of $Tspan7^{-/-}$ rats compared with WT rats (Fig 4C, n = 13, n.s).

The H&E staining showed the gross structure of the hippocampus and decreased pyramidal neuron numbers in CA1, CA2, and CA3 regions of $Tspan7^{-/-}$ rat brains and WT rat brains (Fig 4D–I, n = 6 in the WT group and n = 5 in the $Tspan7^{-/-}$ group, *$P$ < 0.05, **$P$ < 0.01). Further Nissl staining showed that the neurons were more loosely arranged and hypochromic in the $Tspan7^{-/-}$ brain of rats compared with those of WT rats (Fig 4E–J). The Nissl-positive neurons were significantly decreased in CA1, CA2, and CA3 regions of $Tspan7^{-/-}$ brains in comparison with those of WT rats (Fig 4J–M, n = 6 in the WT group and n = 5 in the $Tspan7^{-/-}$ group, *$P$ < 0.05, **$P$ < 0.01), whereas no significant difference in the DG area of the hippocampus and cortex was observed (Fig 4J, N, and O, n = 6 in the WT group and n = 5 in the $Tspan7^{-/-}$ group, n.s).

### TSPAN7 deficiency altered expression of autism-related genes

To further explore the molecular events involved in TSPAN7, we analyzed the expression of nine ASD-related genes by qRT-PCR in the hippocampus and cortex regions of the rats. The results revealed that the expression of four synaptic-related genes of *Ube3a*, *PSD95*, *Nlgn3*, and *Syn* was significantly down-regulated in $Tspan7^{-/-}$ rats (Fig 5A–D, n = 6, *$P$ < 0.05). However, expression of the *Shank3*, *Mecp2*, *Pten*, *Tsc1*, and *Tsc2* was not significant changed in $Tspan7^{-/-}$ rats. (Fig 5E–H, n = 6, n.s). These results suggested that TSPAN7 could be involved in specific autistic features through a synaptic mechanism.

### TSPAN7 deficiency impaired the neuronal morphology

To obtain a complete understanding of the changes in neuronal morphology and dendritic spines in $Tspan7^{-/-}$ rats, we performed Golgi-Cox staining on the hippocampus and cortex of $Tspan7^{-/-}$ rats and WT rats. We found that the dendritic and axonal branching was decreased in the $Tspan7^{-/-}$ brain (Fig 6A, n = 3). Then, the number of intersections of dendrites of neurons at 40–140 $\mu m$ in the $Tspan7^{-/-}$ brain was significantly reduced compared with that of the WT brain (Fig 6B, n = 3, *$P$ < 0.05, ***$P$ < 0.001).

The dendritic spines were divided into five types: filopodia spines, long-thin spines, thin spines, stubby spines, and mushroom spines according to its length and width described by the previous report (Risher et al, 2014). Among them, the length of filopodia was >2 $\mu m$, the length of long-thin spine <2 $\mu m$, the length of thin spine <1 $\mu m$, the ratio of length/width of stubby spine <1, and the width of mushroom spine >0.6 $\mu m$ (Fig 6C). The total number of spines at both the apical dendrites and basal dendrites was significantly reduced in the $Tspan7^{-/-}$ brain compared with that of the WT brain. The reduction of spines mainly existed in mushroom spine, stubby spine, thin spine, and long-thin spine, whereas the filopodia was unchanged obviously (Fig 6D–F, n = 3, *$P$ < 0.05, **$P$ < 0.01, ***$P$ < 0.001, n.s). The integrity of synapses was detected by immunofluorescent staining with PSD95 antibody and SYN antibody. The overlapping puncta of PSD95-immunolabelled postsynaptic and SYN-immunolabelled presynaptic puncta were analyzed, and the results showed that the overlapping puncta were significantly decreased in the hippocampus and cortex of the $Tspan7^{-/-}$ brain compared with the WT brain (Fig 7A–J, n = 3, *$P$ < 0.05).

### TSPAN7 deficiency inactivated the integrin $\beta$1/FAK/SRC signal pathway

The binding of TSPAN7 with integrin $\beta$1 in primary neurons was indicated in previous report (Bassani et al, 2012) and raised a question that how TSPAN7 regulated the integrin $\beta$1 signal pathway. We first observed the decreased levels of integrin $\beta$1 by immunofluorescence staining in primary $Tspan7^{-/-}$ neurons of the hippocampus and cortex. (Fig 8A and C, n = 9, **$P$ < 0.01). A significant reduction of integrin $\beta$1 in total lysates and plasma membrane fraction in the brain tissues of $Tspan7^{-/-}$ rats was then confirmed by Western blot, whereas the levels of integrin $\beta$1 in cytosol fraction were not changed (Fig 8B and D–F, n = 6, **$P$ < 0.01, n.s).

FAK and SRC are the central downstream mediators of the integrin-dependent signal pathway (Kalia et al, 2004; Ribeiro et al, 2013). The phosphorylation of FAK and SRC were also inhibited corresponding to the abnormity of integrin $\beta$1 in the brain tissues of $Tspan7^{-/-}$ rats (Fig 8B, G, and H, n = 6, *$P$ < 0.05). SRC appeared to be involved in regulation of synaptic function (Theus et al, 2006; Cho et al, 2013; Zhang et al, 2018). Corresponding to the inactivation of SRC, the expression of PSD95, SYN, GluR1, and GluR2 was significantly inhibited in brain tissues of $Tspan7^{-/-}$ rats compared with that of WT rats (Fig 8B and I–L, n = 6, **$P$ < 0.01, ***$P$ < 0.001). Our results suggested that the TSPAN7 deficiency, as a binding protein, interrupted integrin $\beta$1 localization on the plasma membrane and then affected the activation FAK and SRC. The inhibition of SRC

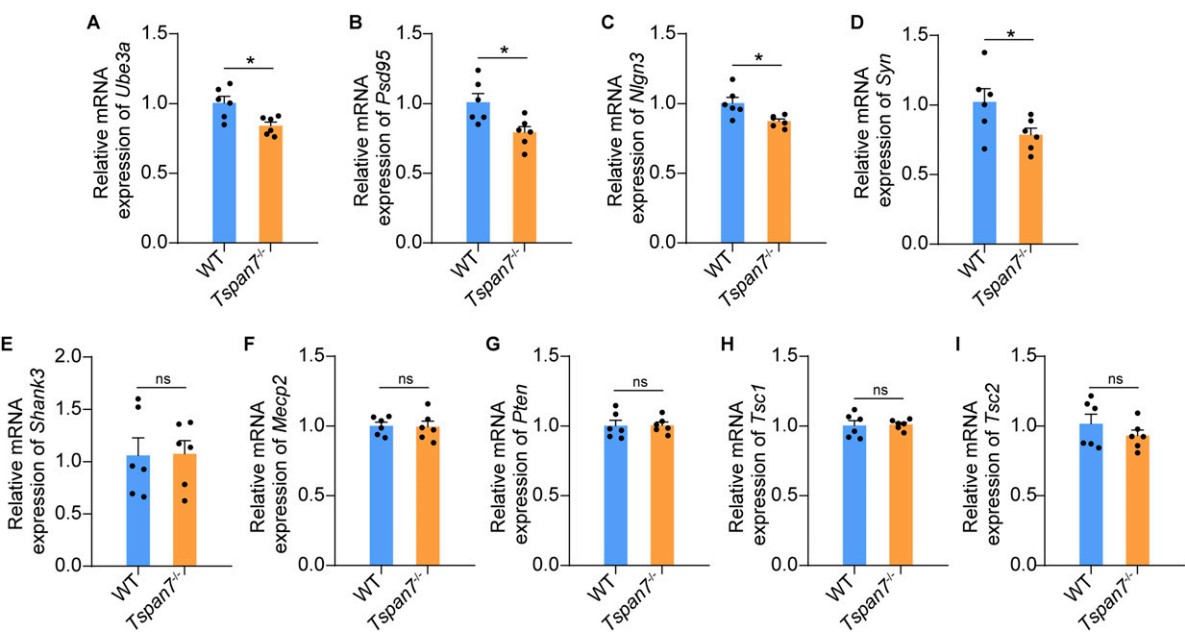

**Figure 5. Determination of the expression of autism-related genes in the *Tspan7^−/−* brain.**
**(A, B, C, D, E, F, G, H, I)** Total RNA was isolated from fresh brains of female WT and *Tspan7^−/−* rats at 6 mo old. The expression of nine autism-related genes was detected by qRT-PCR and normalized to *Gapdh*, including *Ube3a, PSD95, Nlgn3, Syn, Shank3, Mecp2, Pten, Tsc1,* and *Tsc2*. Data information: In (A, B, C, D, E, F, G, H, I), data were analyzed by two-tailed unpaired *t* tests. Error bars represent SEM, n = 6, *P < 0.05, n.s, not significant.
Source data are available for this figure.

could be resulted by the down-regulation of synapse-related genes and impairment of synaptic integrity.

### Recovery of SRC activity improved synaptic damage in primary cultured neurons

We then isolated primary neurons from the brains of WT and *Tspan7^−/−* rats to confirm the results of the roles of SRC in regulating synapse-related genes and synaptic integrity in brain tissues. The WT neurons were treated with or without an SRC inhibitor (bosutinib) and an SRC activator (EPQpYEEIPIYL), and the *Tspan7^−/−* neurons were treated with or without the SRC activator. In WT neurons, when the SRC phosphorylation was inhibited by bosutinib, the expression of PSD95, SYN, GluR1, and GluR2 was significantly inhibited. Like the inhibitor, in *Tspan7^−/−* neurons, *Tspan7* knockout also caused the inhibition of SRC signal and the down-regulation of the synaptic proteins and receptors. But when the SRC activator was administrated to the cells, the SRC phosphorylation, as well as the expression of PSD95, SYN, GluR1, and GluR2, was significantly enhanced or restored in the WT and *Tspan7^−/−* neurons (Fig 9A–F, n = 3, *P < 0.05, **P < 0.01). Furthermore, PSD95 and SYN double immunofluorescence confirmed the impairment effect of SRC inhibitor and *Tspan7* knockout on synaptic structure proteins and the protection effect of SRC activator on synaptic structure proteins in WT and *Tspan7^−/−* neurons (Fig 9G and H, n = 9, *P < 0.05, ***P < 0.001). The results suggested that TSPAN7 knockout caused the impairment of synaptic integrity through regulating the integrin β1/FAK/SRC signal pathway.

## Discussion

*Tspan7* gene mutations are correlated to ID and ASD (Piton et al, 2011; Bassani et al, 2013; Penzes et al, 2013). It has been indicated that TSPAN7 deficiency altered synapse structure and functionally impaired learning and memory (Murru et al, 2017). The summary of data from BlastP, GenBank, PubMed, and Ensembl showed that *Tspan7* was an evolutionarily conserved gene in vertebrates (Fig 1A). The *Tspan7* exhibited a similar expression pattern in human, rats, and mice, which was highly expressed in the brain and lung but lowly in tissues of the spleen, heart, kidney, and liver (Fig 1B, G, and H). *Tspan7* was also found to be down-regulated in patients' brains in a variety of diseases including ASD, HD, PD, and AD (Fig 1C–F, *P < 0.05, **P < 0.01, n.s), implying the important role in brain disease.

Consistent with *Tspan7^−/y* mice, the *Tspan7^−/−* rats showed both of ASD-like and ID-like phenotypes with decreased sociability, increased self-grooming, decreased sucrose preference, and impaired spatial memory and working memory compared with WT rats (Fig 3, n = 13, *P < 0.05, **P < 0.01 ***P < 0.001). Except for the similar behaviors between *Tspan7^−/y* mice and *Tspan7^−/−* rats, we found obvious pathological changes in *Tspan7^−/−* brains, which was not observed in *Tspan7^−/y* mice, including the decreased volume of the hippocampus and cortex (Fig 4A and B, n = 6, *P < 0.05); decreased number of neurons; and Nissl-positive neurons in CA1, CA2, and CA3 regions (Fig 4D–H, n = 5, *P < 0.05, **P < 0.01). The hypochromic hippocampus was also observed in *Tspan7^−/−* brains by Nissl staining (Fig 4E–J), which detected specifically the Nissl bodies (endoplasmic reticulum) in the cytoplasm of neurons and indirectly report the status of protein synthesis in neurons (Kádár et al, 2009).

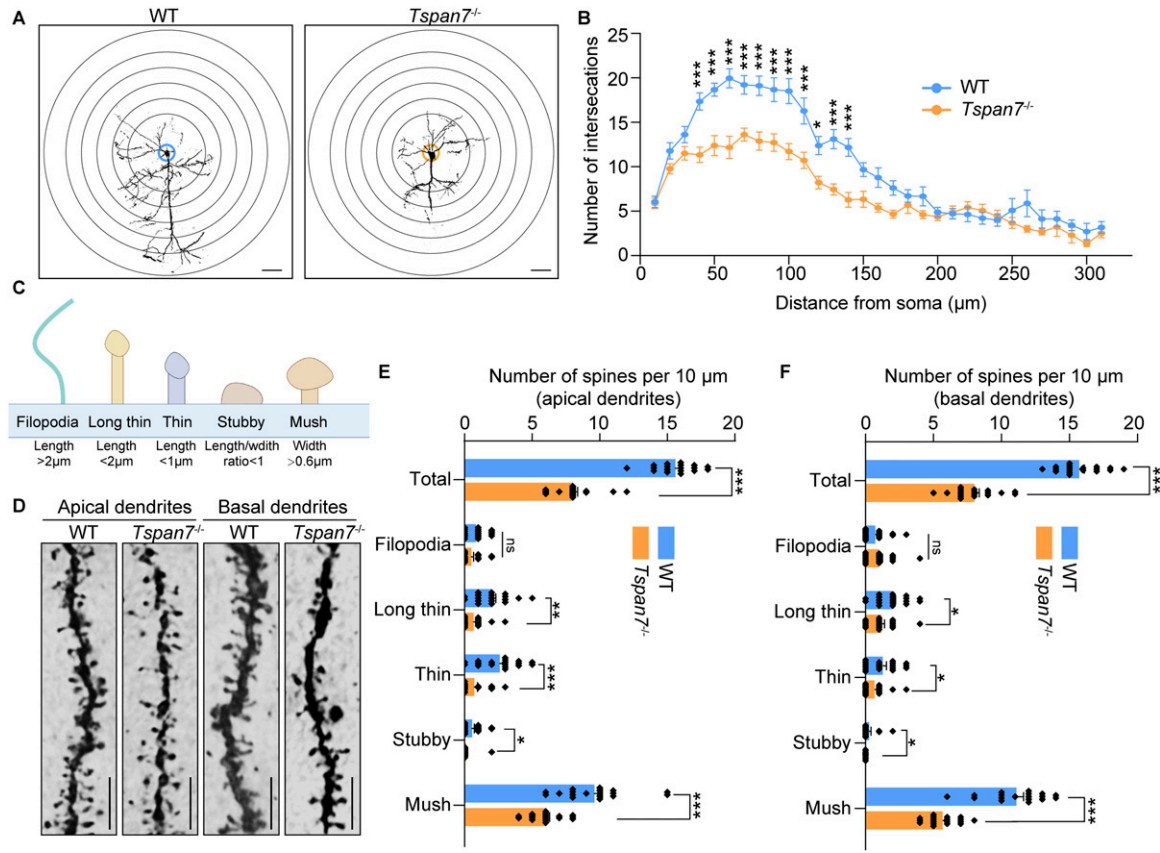

**Figure 6. Analysis of dendritic spines density and neuronal complexity in the brain of the *Tspan7*⁻/⁻ rat.**
**(A)** Coronal sections of brains from female WT and *Tspan7*⁻/⁻ rats at 6 mo old were stained by Golgi-Cox staining. Representative photomicrographs of neurons with the allocation of dendrites between repeated concentric rings were showed; scale bar = 50 µm. **(B)** Number of neuronal intersections that indicated neuronal complexity was analyzed using the Sholl analysis tool of ImageJ. **(C)** Standard of dendritic spines classification used in this study and the morphology schematic diagram of five different types of dendritic spines: filopodia, long thin, thin, stubby, and mushroom spines. **(D)** Representative images of apical and basal dendrites in the neurons from WT and *Tspan7*⁻/⁻ rats. Scale bar = 10 µm. **(E, F)** Total number of dendrite spines and the numbers of five different types of dendrite spines on apical or basal dendrites from WT and *Tspan7*⁻/⁻ rats were quantified using Reconstruct software. Data information: In (B), data were analyzed by two-way ANOVA with Sidak's multiple comparison method. In (E, F), data were analyzed by two-tailed unpaired $t$ tests. Error bars represent SEM, n = 3. Each point represents data from one spine, and at least six spines were analyzed for each rat, *$P < 0.05$, **$P < 0.01$, ***$P < 0.001$, n.s, not significant. Diagram of (C) was created with BioRender.com. Source data are available for this figure.

These results suggested that the TSPAN7 deficiency resulted in the weakness of hippocampal neurons compared with that of WT rats. Furthermore, we detected expression of several ASD-related genes in rat brains, which was involved in various biological processes, such as an abnormal mTOR signaling pathway, cell growth, cell proliferation, and chromatin remodeling (Kwon et al, 2006; Tsai et al, 2012; Yin & Schaaf, 2017).

Previous studies found that total brain weight and cortical volume were significantly reduced in Ube3a- and NLGN3-deficient mice (Varghese et al 2017). In the *Tspan7*⁻/⁻ rat model, we found that the volume of the cortex and hippocampus of the *Tspan7*⁻/⁻ rats was significantly reduced. The expression of Ube3a, PSD95, Nlgn3, and Syn were significantly down-regulated in the *Tspan7*⁻/⁻ hippocampus and cortex compared with those of WT rats (Fig 5A–D, n = 6, *$P < 0.05$), which are important for synapse formation and remodeling (Kim et al, 2008; Liu et al, 2018; Xu et al, 2018; Chen et al, 2020). It suggested that the down-regulation of synapse-associated genes such as *Ube3a, PSD95, Nlgn3*, and *Syn* in the hippocampus and cortex was closely associated with the decrease in hippocampal and cortical volume caused by

TSPAN7 knockout, and TSPAN7 deficiency could be involved in pathogenesis of ASD through a synaptic mechanism.

We then analyzed the changes of neuron morphology and dendritic spines and found that neuronal complexity and dendritic spines were significantly decreased in the *Tspan7*⁻/⁻ brain compared with the WT brain (Fig 6, n = 3, *$P < 0.05$ **$P < 0.01$, ***$P < 0.001$).

TSPAN7 is the member of the tetraspannin family, which regulated proteins interacting with C kinase 1 (PICK1), and the PICK1 guides the AMPA receptor trafficking by its PDZ domain in primary neurons (Bassani et al, 2012). However, activation of PKC- ERK, rather than the changes of PICK1, PSD95, and SYN, was detected in *Tspan7*⁻/ʸ mice (Murru et al, 2017, 2021). Unexpectedly, we did not observe any significant changes of phosphorylation on ERK, PKCε (PKC epsilon), and PKCα and expression levels of ERK, PKCε, PKCα, PICK1, and NMDAR1 in *Tspan7*⁻/⁻ brains compared with those of WT rats (Fig S1, n = 5).

Besides the PICK1, integrin β1 has been identified as another binding protein of TSPAN7 (Bassani et al, 2012), suggesting that TSPAN7 might be involved in integrin β1–initiated signaling. We found impaired integrin β1 distribution in primary neurons of

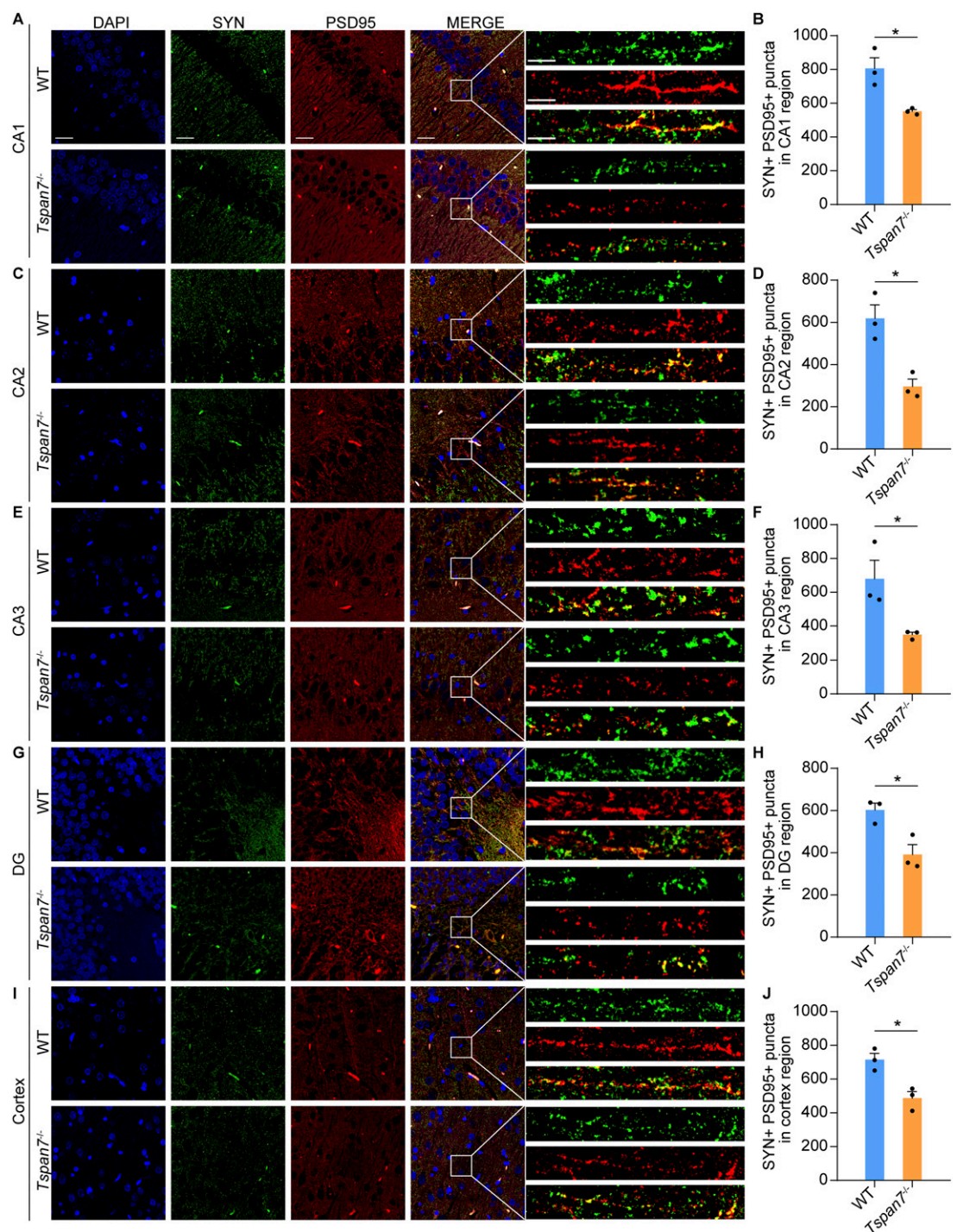

**Figure 7. Analysis of the synaptic integrity in the rat brain section.**
**(A, C, E, G, I)** Brain sections from WT and $Tspan7^{-/-}$ rats were stained with the PSD95 antibody (red) and SYN antibody (green). The integrity of synapses were observed by the overlapped puncta immunolabelled by the postsynaptic marker PSD95 and the presynaptic marker in the hippocampus (CA1, CA2, CA3, and DG) and the cortex of WT and $Tspan7^{-/-}$ brains. **(B, D, F, H, J)** Number of puncta overlapping PSD95-immunolabelled postsynaptic puncta was counted to quantity synaptic integrity in the brains of WT and $Tspan7^{-/-}$ rats. Data information: In (B, D, F, H, J), data were analyzed by two-tailed unpaired $t$ tests. Error bars represent SEM, n = 3, *$P < 0.05$. Scale bar = 25 $\mu$m, scale bar = 5 $\mu$m.
Source data are available for this figure.

$Tspan7^{-/-}$ rats (Fig 8A and C, n = 9, **$P < 0.01$) and reduction of integrin $\beta$1 in the plasma membrane fraction of the $Tspan7^{-/-}$ brain tissues (Fig 8B, D, and E, n = 5, *$P < 0.05$. n.s). The result indicated that the cell membrane localization of integrin $\beta$1 was

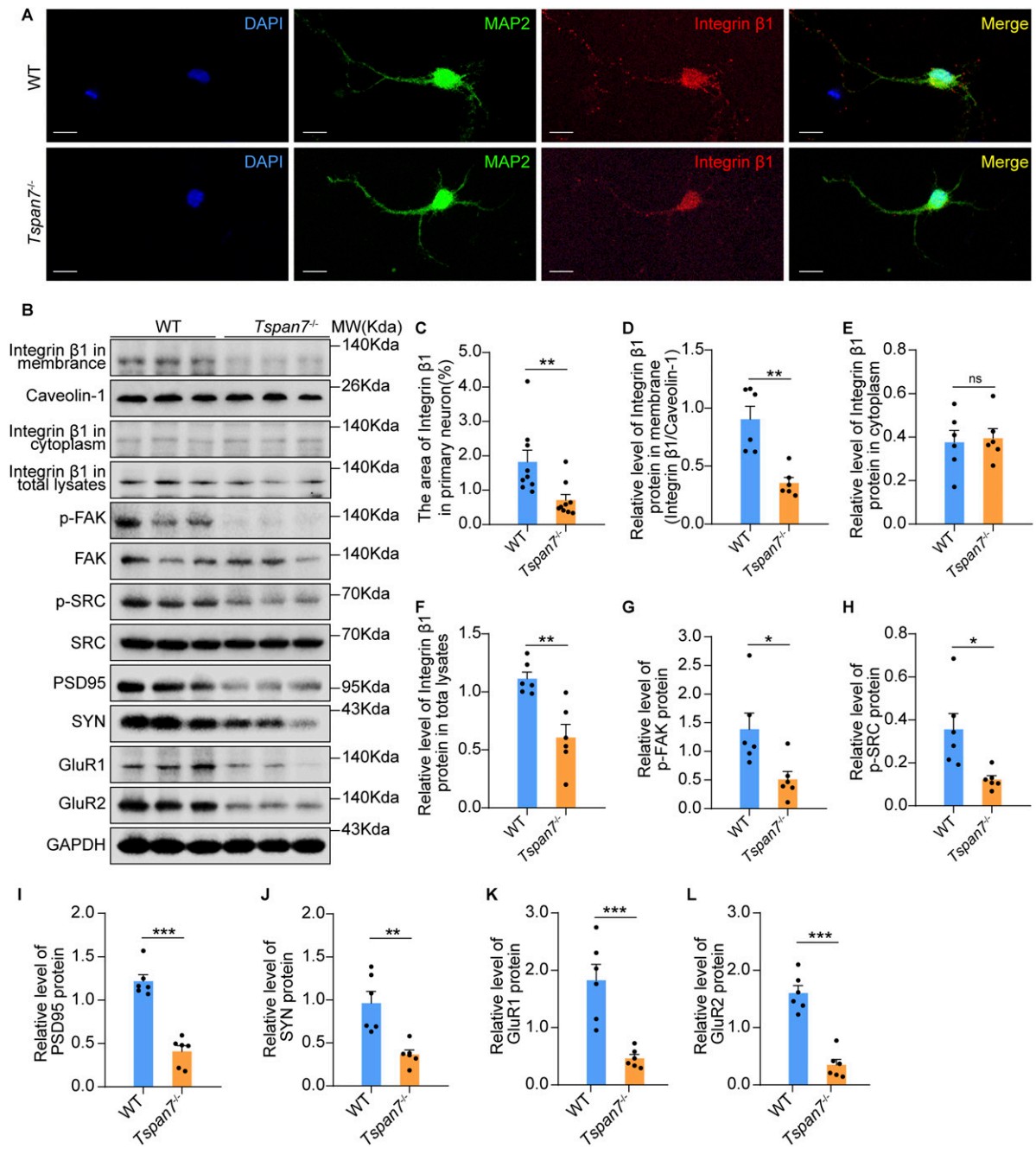

**Figure 8. Determination of the signal pathways in brain tissues.**
**(A)** Primary neurons from embryonic day 17. WT rats and *Tspan7⁻/⁻* rats were cultured for 6 d and then were stained by double immunofluorescence with antibodies of anti-MAP2 (green) and anti-integrin *β*1 (red). Scale bar = 10 *μ*m. **(B)** Plasma membrane fractions, cytosol fractions, and total lysates of brain tissues from female WT and *Tspan7⁻/⁻* rats at 6 mo old were prepared and separated on SDS–PAGE, respectively. Integrin *β*1 in the membrane, cytosol fractions, and total lysates and phosphorylated and total FAK, SRC, PSD95, SYN, GluR1, and GluR2 in total lysates were detected with Western blot. **(C)** Integrin *β*1–immunolabelled area in primary neurons was quantified. **(D, E, F, G, H, I, J, K, L)** Expression levels of integrin *β*1 in the membrane, cytosol fractions, and total lysates and phosphorylated FAK, SRC, PSD95, SYN, GluR1, and GluR2 in total lysates were quantified with a software of ImageJ. n = 6. Data information: In (C, D, E, F, G, H, I, J, K), data were analyzed by two-tailed unpaired *t* tests. Error bars represent SEM, *P < 0.05, **P < 0.01, ***P < 0.001, n.s, not significant. **(C)** Each point represents data from one cell in (C), and at least nine cells were analyzed. Source data are available for this figure.

interrupted by TSPAN7 deficiency. The phosphorylation of its downstream FAK and SRC proteins was sequentially decreased (Fig 8B, F, and G, n = 5, *P < 0.05), and the expression of PSD95, SYN, and GluR1/2 was significantly inhibited in brain tissues of

*Tspan7⁻/⁻* rats compared with those of WT rats (Fig 8B and H–K, n = 5, **P < 0.01, ***P < 0.001).

Integrin *β*1 is localized to the excitatory-synapse postsynaptic membrane and is critical for synaptic integrity (Mortillo et al, 2012;

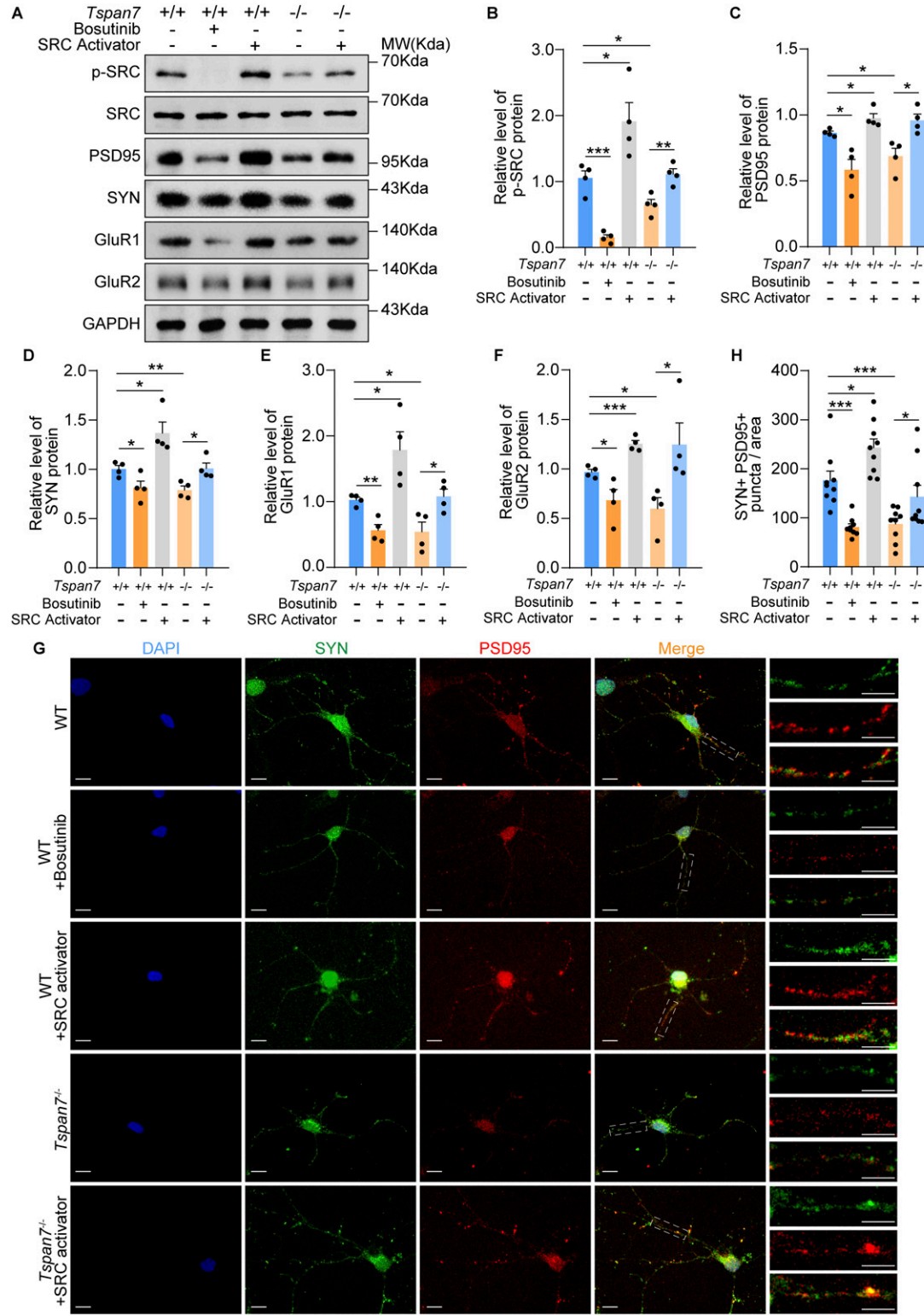

**Figure 9. Determination of the signal pathways in primary cultured rat neurons.**
**(A)** Primary neurons from embryo brains of of WT rats and $Tspan7^{-/-}$ rats at E17 were cultured for 6 d. The WT neurons were treated without or with 10 $\mu$M bosutinib for 4 h or 10 $\mu$M Src activator for 1 h. The $Tspan7^{-/-}$ neurons were treated without or with 10 $\mu$M SRC activator for 1 h. Then phosphorylated and total SRC, PSD95, SYN, GluR1, and GluR2 in the primary neurons were detected with Western blot. **(B, C, D, E, F)** Expression levels of phosphorylated SRC, PSD95, SYN, GluR1, and GluR2 from primary neurons were quantified with ImageJ. n = 4, *P < 0.05, **P < 0.01, ***P < 0.001. **(G)** Primary neurons were stained by double immunofluorescence with antibodies of anti-SYN (green) and anti-PSD95 (red) to observe the overlapping puncta of PSD95-immunolabelled postsynaptic puncta and SYN-immunolabelled presynaptic puncta in the WT and

Heintz et al, 2014; Wang et al, 2016). The activated or clustered integrin β1 in the cell membrane recruits cytoskeletal proteins, RhoGTPases, and non-receptor tyrosine kinases, including FAK and SRC (Mitra et al, 2005). The recruited FAK is autophosphorylated at tyrosine 397 and then causes SRC recruitment and activation (Arold, 2011). FAK is a major protein that integrates integrin β1 signaling and function in neurons (Ribeiro et al, 2013). SRC is widely expressed in neurons and implicated in proliferation and differentiation during the development of the central nervous system (Kalia et al, 2004). SRC is also a crucial element and appears to be involved in regulation of synaptic integrity, which can indirectly regulate the expression of PSD95 (Cho et al, 2013), glutamate receptors (Zhang et al, 2018), and synaptophysin (Theus et al, 2006).

Our results showed that inhibition of SRC phosphorylation by bosutinib reduced the expression of PSD95, SYN, and GluR1/2 and impaired synaptic integrity in primary WT neurons, whereas the level of synaptic proteins and glutamate receptors were enhanced by the recovery of SRC activity with an activator in $Tspan7^{-/-}$ neurons (Fig 9, n = 9, *$P$ < 0.05, ***$P$ < 0.001). The results suggested that SRC was essential for regulation of PSD95, SYN, and GluR1/2 expression via TSPAN7/integrin β1/FAK signaling.

In conclusion, our finding indicated that TSPAN7 deficiency caused both of ASD-like and ID-like phenotypes in rats and obvious pathological changes in brain tissues including neuron loss and synaptic impairment in the hippocampus. The integrin β1/FAK/SRC signal pathway was interrupted by TSPAN7 deficiency and resulted in the down-regulation of PSD95, SYN, and GluR1/2, which were essential proteins of synaptic integrity. Our results suggested that TSPAN7/integrin β1/FAK/SRC was a new signal pathway involved in pathogenesis of ASD and ID related to TSPAN7.

# Materials and Methods

## Data extraction

The TSPAN7 protein sequence of different species was obtained from the GenBank database. Multiple sequence alignment of TSPAN7 sequences of human (*H. sapiens*; NP_004606), macaque (*M. mulatta*; NP_001248645), rat (*R. norvegicus*; NP_001385666), mouse (*M. musculus*; NP_062608), clawed frog (*X. tropicalis*; NP_989350), and zebrafish (*D. rerio*; NP_999939) was performed by tools like DNAMAN and Clustal, and the identities of the amino acid sequence were analyzed. The expression of TSPAN7 in different tissues in human, rats, and mice was obtained from the GenBank database (https://www.ncbi.nlm.nih.gov/gene/7102, https://www.ncbi.nlm.nih.gov/gene/363447, https://www.ncbi.nlm.nih.gov/gene/21912). The different expression data of TSPAN7 in human diseases were obtained from the NCBI Gene Expression Omnibus database. "Homo sapiens" AND "tissue" AND "disease" were first searched in the Gene Expression Omnibus datasets, and then the data with small sample sizes (n < 3)

or no control group were filtered out. Finally, the data were identified for analysis, including ASD (GSE30573), HD (GSE129473), AD (GSE36980), and PD (GSE20295).

## Animals

The *Tspan7* knockout rats were generated using CRISPR/Cas9 following our previously described procedure (Ma et al, 2016). Briefly, sgRNA (CCACCATGGTGGGCATGTACT) and sgRNA (CCTGGCACAGTAAACAGAGTC) were designed to target exon 2 of *Tspan7*. The mixture containing Cas9 protein (30 ng/μl) and sgRNAs (10 ng/μl each) was microinjected into the cytoplasm and male pronucleus of the fertilized eggs from Wistar rats. The microinjected eggs were then transferred to pseudopregnant Wistar rats who gave birth to pups. The litters were genotyped by PCR with primers of 5'-GCAAACTACCACATCATTCCCAG-3' and 5'-CTGTGTCCATCAGCTCAGATCAG-3'. The PCR fragments were analyzed by DNA sequencing to confirm the knockout DNA fragment. All rats used in this study were housed in groups and bred in an AAALAC-accredited facility, and all rat experiments were approved by the Animal Care and Use Committees of the Institute of Laboratory Animal Science of Peking Union Medical College (Approval no. MYW22004).

## Quantitative real-time PCR

The fresh hippocampal and cortical tissues were sampled from WT female Wistar rats (WT) and female *Tspan7* knockout rats ($Tspan7^{-/-}$). Total RNA in brain tissues was isolated by TRIzol reagent (15596018; Invitrogen). The cDNA was produced with a reverse-transcription kit (RR047A; Takara). Quantitative real-time PCR (qRT-PCR) was applyed using SYBGreen real-time PCR kits (RR820A; Takara) on a PCR equipment (QuanStudioTM3 Real-Time PCR instrument; Thermo Fisher Scientific). The mRNA expression of nine genes related with autism was detected and normalized to *Gapdh*, including *Tspan7*, *Ube3a*, *PSD95*, *Nlgn3*, *Shank3*, *Mecp2*, *Pten*, *Tsc1*, and *Tsc2*. The primers for the quantitative real-time PCR are summarized in Table 1.

## MRI

The MRI examination was performed on an 7.0 T small-animal MRI system (Varian). During imaging, WT and $Tspan7^{-/-}$ rats (n = 6/group) were anesthetized with medical oxygen (1 Liter/min) containing 2% isoflurane. T2-weighted MRI images in the coronal plane were acquired with a fast spin echo (fsems) sequence with following parameters: FOV (field of view) = 35 × 35 mm; no slices = 20; slice thickness = 1 mm (zero slice gap); TR (repetition time) = 3,500 ms; TE (echo time) = 18.0 ms; image matrix = 256 × 256; and number of averages = 10. Volumes (region of the cortex, hippocampus, cerebellum, and cerebrum of WT and $Tspan7^{-/-}$ rats) were then drawn on the T2 images to determine the change of volumes (Vnmrj; Varian).

---

$Tspan7^{-/-}$ neurons. Scale bar = 10 μm, scale bar = 5 μm. **(H)** Number of puncta overlapping PSD95-immunolabelled postsynaptic puncta in the WT and $Tspan7^{-/-}$ neurons were counted to detect the integrity of synapse. n = 9, *$P$ < 0.05, ***$P$ < 0.001. Data information: In (B, C, D, E, F, G), data were analyzed by two-tailed unpaired $t$ tests. Error bars represent SEM.
Source data are available for this figure.

**Table 1. Primers used in this study.**

| Gene | Forward sequences | Reverse sequences |
|------|-------------------|-------------------|
| Ube3a | 5′-GGGGCGAGGACAGGTTAAAAA-3′ | 5′-TGGCCATTCGGTGACATCAG-3′ |
| PSD95 | 5′-TCATAACTCCCCATGCCATT-3′ | 5′-CTCATGCAAACCAGCAAAGA-3′ |
| Nlgn3 | 5′-GCCCACGGAAGATGTAAAGC-3′ | 5′-GCTAAGTCCTCGCCCTGTTT-3′ |
| Syn | 5′-CCACGGACCCAGAGAACATT-3′ | 5′-TTCAGGAAGCCAAACACCACT-3′ |
| Shank3 | 5′-GGCCTGAGGATGACAAACCA-3′ | 5′-CTGCTGAAGAGCCGAGCTG-3′ |
| Mecp2 | 5′-GCGACGTTCCATCATTCGTG-3′ | 5′-GGCTTTTCCCTGGGGATTGA-3′ |
| Pten | 5′-AGGACCAGAGATAAAAAGGGAGT-3′ | 5′-CCTTTAGCTGGCAGACCACA-3′ |
| Tsc1 | 5′-AGTGACCGCGGATTAGAGGA-3′ | 5′-GAAGGGAGAGTCAAAGCCCC-3′ |
| Tsc2 | 5′-CGCGCGGGAGCAGTT-3′ | 5′-GCACACCTAGGATTTGGCCT-3′ |
| Gapdh | 5′-CTCATGACCACAGTCCATGC-3′ | 5′-TTCAGCTCTGGGATGACCTT-3′ |

## Behavior analysis

The female WT and $Tspan7^{-/-}$ rats at 6 mo old were applied to five kinds of behavior analysis to estimate the levels of autism and cognitive damage of $Tspan7^{-/-}$ rats.

For grooming behavior, the procedure was performed as previously described (Murru et al, 2021). Rats were individually placed into an open-field arena (80 × 80 cm) and recorded for self-grooming behavior. The experiment was recorded for a total of 20 min, and the total time spent grooming and the total number of times the rats were groomed during the last 10 min were measured. Self-grooming was defined as licking paws, washing the nose and face, or scratching fur with any foot.

The sucrose preference test was performed according to a previous study (Schiavi et al, 2019). Rats were given two bottles, one of sucrose water (1%) and one of tap water. The amount of sucrose water and pure water consumed was evaluated at every 24 h. The preference for the sucrose solution was calculated as the percentage of sucrose solution ingested relative to the total amount of liquid consumed.

Social interaction analysis was performed according to the reported methods (Murru et al, 2021). A stranger rat (stimulus rat) and an object are, respectively, placed inside a cage in the left or right compartment. The testing rat was placed in the middle compartment and let it explore around randomly. The time of the testing rat spent on the stimulus rat and object was monitored for 10 min. The discrimination index was measured as the difference in time spent with the stimulus rat and the objective, expressed as the percentage ratio of the total time spent exploring both the stimulus animal and the objective.

The Y-maze test was performed according to reported methods (Li et al, 2020), in a rat Y-maze apparatus (arm length 90 cm). Briefly, after one rat was placed at the end of one arm, the animal was allowed to explore freely in the three arms of the maze for 5 min. The activity of the rats was acquired and analyzed using a Visu Track system (XR-VT; xinruan). The sequence number of arm entries was recorded to calculate the percentage of alternation.

The Morris water maze test was modified from a protocol of our previous study (Pang et al, 2020). Briefly, the rats were placed in a large round pool with a radius of 75 cm, and the animals were allowed to swim for 60 s once and trained to find the hidden platform twice 1 d for five straight days. The time to find the platform was recorded as latency. On day 6, the rats were placed in a contralateral quadrant to swim for 120 s in the maze without the platform. The platform crossing number and target quadrant duration were recorded using a video-tracking system as above.

## Histological analysis

The brain tissues were first fixed in formalin (10%) for 2 d at room temperature, and the 4-μm thick coronal paraffin sections were prepared as a standard histological procedure.

The dewaxed and rehydrated sections were first stained with H&E as previously described (Lu et al, 2018). The stained sections were then scanned under a digital slide scanner (Pannoramic 250 FLASH; 3DHISTECH), and the images were captured by CaseViewer. The images were quantified using ImageJ software. For Nissl staining, the dewaxed and rehydrated coronal sections were stained in toluidine blue solution (G1436; Solarbio) for 30 min at 55°C. After clearing in distilled water, the slides were gradually dehydrated and given dimethylbenzene. Then the slides were coverslipped with neutral balsam. Finally, the numbers of surviving neurons per ×400 field within the hippocampal CA1 were counted.

For immunofluorescent staining, the growing adherent cells on coverslips were fixed in paraformaldehyde (4%) for 20 min at room temperature and penetrated with 0.1% Triton X-100 for 5 min. The coverslips or the dewaxed and rehydrated sections were applied to immunofluorescent staining as previously described (Li et al, 2012; Dong et al, 2021). Briefly, the primary neurons were fixed in paraformaldehyde (4%) for 20 min at room temperature and penetrated with 0.1% Triton X-100 for 5 min. Then the slices washed 3 times with PBS for 5 min. The sections were blocked in 1% BSA at room temperature for 1 h and then incubated with primary antibodies overnight at 4°C. The sections were incubated with secondary fluorescent antibodies for 1 h in a dark box at room temperature and then washed off by PBS. The sections were mounted with DAPI buffer (ZLI-9557; ZSGB BIO) and observed by two-photon confocal laser scanning microscopy (Leica). The images were quantified using ImageJ software. Images were first converted to 8-bit gray scale and binary threshold to highlight a positive staining. Finally,

**Table 2. Antibodies used in this study.**

| Primary antibodies | Source | Catalog no. | WB | IF |
|---|---|---|---|---|
| TSPAN7 | Thermo Fisher Scientific | PA5-76938 | 1:500 | 1:150 |
| Integrin $\beta$1 | Abcam | ab95623 | 1:500 | 1:150 |
| Caveolin-1 | CST | 3267 | 1:500 | — |
| p-FAK | Abcam | ab81298 | 1:1,000 | 1:200 |
| FAK | ProteinTech | 66258-1-Ig | 1:1,000 | — |
| p-SRC | CST | 2101 | 1:500 | — |
| SRC | ProteinTech | 11994 | 1:500 | — |
| PSD95 | Millipore | MAB1596 | 1:2,000 | 1:100 |
| SYN | ProteinTech | 17785-AP | 1:500 | 1:200 |
| GluR1 | Abcam | ab51092 | 1:1,000 | — |
| GluR2 | ProteinTech | 11994 | 1:1,000 | — |
| NEUN | Abcam | ab104224 | — | 1:75 |
| NMDAR1 | Abcam | ab17345 | 1:500 | |
| GAPDH | Abcam | ab201822 | 1:10,000 | — |
| 488 Goat anti- rabbit IgG | Invitrogen | A11034 | — | 1:200 |
| 555 Goat anti- mouse IgG | Invitrogen | A21424 | — | 1:200 |
| HRP-labelled goat anti-mouse IgG | ZSGB BIO | ZB-2305 | 1:10,000 | |
| HRP-labelled goat anti-rabbit IgG | ZSGB BIO | ZB-2301 | 1:10,000 | |
| HRP-labelled goat anti-rat IgG | ZSGB BIO | ZB-2307 | 1:10,000 | |

the percentage of the area positive for integrin-$\beta$1 was measured by ImageJ.

The primary antibodies and secondary antibodies used in this work were listed as Table 2.

## Golgi-Cox staining

The manufacturer's instructions were strictly followed (FD Rapid GolgiStain Kit; PK401; FD Neurotechnologies, INC.). The fresh hippocampal and cortical tissues were rinsed with double-distilled water. The tissues were sequentially placed in a mixture of solutions A and B and solution C in the dark in the certain time. Frozen coronal sections of 90-$\mu$m thickness were prepared on a freezing microtome (POLAR-D-JC; SAKURA SEIKI) at −22 °C and transferred to microscope slides onto a few drops of solution C. The slides were dried at RT in the dark overnight. Serial sections were stained with mixture of D and E solutions for 10 min, then dehydrated, cleared, and covered with a glass coverslip using neutral resin. Single-plane images were captured using a digital slide scanner (Pannoramic 250 FLASH; 3DHISTECH).

## Quantification of dendritic spines

Images of shole neurons were obtained six cells per brain slice at ×20 magnification. Two parameters (dendritic complexity and dendritic spines) were analyzed. First, dendritic complexity was analyzed by the concentric circle method reported by Sholl using ImageJ (Sholl & Uttley, 1953). Then the dendritic spines were analyzed as previously mentioned (Shibata et al, 2021). Briefly, nearly

20 images could be obtained per z-stack to cover the entire dendrite thickness. The spine counts were analyzed by Reconstruct 1.1.0.0. software, which is publicly available at http://www.bu.edu/neural/Reconstruct.html. We had divided the entire dendrite into 10-$\mu$m segments and calculated the number of spines, which were traced for length and width. The length and width ratios were used to define the spine subtype as previously described (Risher et al, 2014).

## Immunoblot

For protein assay, the brain tissues were, respectively, homogenized in lysis buffer containing RIPA buffer (50 mM Tris; 150 mM NaCl; 1% Triton X-100; 1% sodium deoxycholate; 0.1% SDS, P0013B; Beyotime), a protease inhibitor mixture (87785; Thermo Fisher Scientific), a phosphatase inhibitor cocktail (78420; Thermo Fisher Scientific), and 1 mM phenylmethyl-sulfonylfluoride (PMSF; 36978; Thermo Fisher Scientific) as previously reported (Pang et al, 2021). The proteins in the cell membrane fraction and cytoplasm fraction were extracted using a ProteoExtract Subcellular Proteome Extraction Kit (539790; CALBIOCHEM). Then samples were, respectively, separated on a SDS–PAGE, and the proteins were electroeluted onto a nitrocellulose membrane (Immobilon NC; Millipore). Immunoblots were incubated in primary antibodies and then incubated in HRP-linked secondary antibodies for 1 h. Proteins were visualized using a chemiluminescent detection system (Western Blotting Luminal Reagent; Santa Cruz Biotechnology, Inc.) and analyzed using Quantity One software (version 3.0; Bio-Rad Laboratories, Inc.). The antibodies and working dilution are summarized in Table 2.

### Isolation and primary culture of rat neurons

Dissociated primary brain cultures were performed using a modified protocol (Kaech & Banker, 2006). Briefly, the embryonic hippocampal and cortical tissues at embryonic day 17 (E17) from WT and *Tspan7*$^{-/-}$ rats were dissected and dissociated in 2 mg/ml papain (G8430; Solarbio) and 1% DNase (2270A; Takara). Dissociated cells were suspended in DMEM-F-12 medium (2276628; Gibco) with FBS (10099-141C; Gibco) and then plated at a density of 200,000 cells/cm$^2$ in a six-well plate coated with poly-D-lysin (D6790; Solarbio) or at 50,000 cells/cm$^2$ in a 12-well plate within coverslips coated with poly-D-lysin-coated. 4 h after plating, the DMEM-F-12 medium was changed to neurobasal medium containing 2% B27 (2450340; Gibco), 0.5 Mm L-GlutaMAX (35050061; Gibco), and 0.05 mg/ml antibiotic penicillin streptomycin (15070063; Gibco). The cells were cultured for 6 d. The WT neurons were then treated with or without 10 $\mu$M SRC inhibitor (Bosutinib HY-10158; MedChemExpress) for 4 h. As previously reported (Park et al, 2015), the WT *Tspan7*$^{-/-}$ neurons were starved using DMEM-F-12 medium supplemented with 0.1% BSA for 2 h followed by stimulation with serum or a 10 $\mu$M Src activator (EPQpYEEIPIYL; HY-P3279; MedChemExpress) for 1 h. The cells in the plate were sampled in lysis buffer for Western blot, and the cells on coverslips were fixed in paraformaldehyde (4%) at room temperature for immunofluorescent staining.

### Quantification of postsynaptic and presynaptic puncta marked by immunostaining

The postsynaptic and presynaptic puncta were analyzed as previously described (Ippolito & Eroglu, 2010; Block et al, 2022). For each area of the brain, using both the 488-nm channel to detect SYN and the 555-nm channel to detect the PSD95, a z-stack for each SC section was collected in LAS X software for a total depth of 5 $\mu$m (15 × 0.33 $\mu$m optical sections) at 63× magnification using multi-photon laser scanning microscopy. Maximum image projections were generated for groups of three consecutive optical sections yielding five maximum image projections/section each representing 1 $\mu$m of depth. Maximum projections of three consecutive optical sections were generated using ImageJ. The Puncta Analyzer Plugin (available at: https://doi.org/10.5281/zenodo.6800214) for ImageJ was used to count the number of colocalized synaptic puncta and the number of synaptic puncta produced by Puncta Analyzer.

### Statistical analysis

The data were analyzed using two-tailed unpaired $t$ tests and two-way ANOVA with Sidak's multiple comparison method. Data are presented as mean ± SEM. $P < 0.05$ was considered to indicate a statistically significant difference.

## Supplementary Information

## Acknowledgements

This study was supported by the National Key Research and Development Program of China (2022YFF0710702), the CAMS Innovation Fund for Medical Sciences (CIFMS, 2021-I2M-1-034), and the National Natural Science Foundation of China (grant no. 31970508, 31900380).

## Author Contributions

S Pang: software, formal analysis, visualization, methodology, and writing—original draft.
Z Luo: resources, validation, and methodology.
W Dong: resources, software, and methodology.
S Gao: resources and validation.
W Chen: resources.
N Liu: resources.
X Zhang: resources.
X Gao: resources.
J Li: resources.
K Gao: software and methodology.
X Shi: software and methodology.
F Guan: resources, data curation, and validation.
L Zhang: data curation, supervision, funding acquisition, and writing—review and editing.
L Zhang: resources, supervision, funding acquisition, project administration, and writing—review and editing.

## Conflict of Interest Statement

The authors declare that they have no conflict of interest.

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
