## [Reviewer comments · Life Science Alliance]

Life Science Alliance

Integrin β 1/FAK/SRC signal pathway is involved in autism spectrum disorder in Tspan7 knockout rats

Shuo Pang, Zhuo Luo, Wei Dong, Shan Gao, Wei Chen, Ning Liu, Xu Zhang, Xiang Gao, Jing Li, Kai Gao, Xu Shi, Fei Guan, Li Zhang, and Lianfeng Zhang

DOI: <https://doi.org/10.26508/lsa.202201616>

Corresponding author(s): Lianfeng Zhang, Chinese Academy of Medical Sciences & Peking Union Medical College and Li Zhang, Peking Union Medical College, Chinese Academy of Medical Sciences

Review Timeline:

Submission Date:	2022-07-20
Editorial Decision:	2022-10-05
Revision Received:	2022-11-16
Editorial Decision:	2022-12-01
Revision Received:	2022-12-02
Accepted:	2022-12-02

Scientific Editor: Novella Guidi

Transaction Report:

October 5, 2022

Re: Life Science Alliance manuscript #LSA-2022-01616-T

Professor Lianfeng Zhang
Institute of Laboratory Animal Science, Chinese Academy of Medical Sciences
Key Laboratory of Human Disease Comparative Medicine, Ministry of Health
#5 Panjiayuan Nanli, Chaoyang District, Beijing, China
beijing 100021
China

Dear Dr. Zhang,

Thank you for submitting your manuscript entitled "Integrin β 1/FAK/SRC signal pathway is involved in autism spectrum disorder in Tspan7 knockout rats" to Life Science Alliance. The manuscript was assessed by expert reviewers, whose comments are appended to this letter. We invite you to submit a revised manuscript addressing the Reviewer comments.

Thank you for this interesting contribution to Life Science Alliance. We are looking forward to receiving your revised manuscript.

Sincerely,

B. MANUSCRIPT ORGANIZATION AND FORMATTING:

Reviewer #1 (Comments to the Authors (Required)):

In this manuscript, Pang and colleagues examine the behavioral consequences and underlying mechanism of deletion of the autism-associated gene TSPAN7 in rats. First, they show that knockdown of Tspan7 affects multiple behavioral parameters including self-grooming, social preference, and spatial memory. Secondly, by performing MRI and histological analyses, they show that Tspan7 deletion affects total volume, cell number, and spine density in the cortex and hippocampus. Finally, they show that in primary cultured neurons Tspan7 deletion alters the Integrin beta-1 signaling pathway, and suggest this as a possible mechanism for some of the phenotypes associated with Tspan7 deletion. The manuscript improves the understanding of the behavioral consequences and neurobiological mechanism of Tspan7 deletion and provides interesting data on a potential novel mechanism of how Tspan7 affects synaptic function. However, I have some concerns regarding the data interpretation and analysis, as well as some questions and comments that I hope will improve the manuscript.

Major comments:

1. Some blots and immunostainings lack statistical support for the claims made in the text. The text states that Tspan7 expression is the highest in the rat brain, but there is no quantification for the supporting data in Figure 1G-H. As the blot itself is unconvincing with no clear Tspan7 band and uneven GAPDH loading for most tissues, providing quantification is essential to support this claim. Similarly, no quantification is provided for the stated altered morphology of neurons shown in Figure 4D.

Furthermore, some of the quantifications shown do not support the conclusion drawn in the text. For example, the authors write that Tspan7 expression increases from 0-2 months and then decreases from 2-4 months of age. However, the data in Figure 2b-c does not support this claim: no change is statistically significant and the spread is too large to draw any clear conclusions. On page 7, the authors write that the brain coefficient is slightly decreased compared to WT rats, but the data in Figure 4c show an equal distribution, not at all supporting this claim. The authors should correct these statements.

2. More clarity on which brain regions the data is generated from would be helpful for several figures. Is the data in Figure 5 from the whole brain? If so, why was this chosen instead of for example hippocampus or cortex where the volume and histological differences were identified? I have the same question for the data from Figure 6. What brain regions are these neurons from, and were the brain regions matched closely between wildtype and knockout rats? Since regional effects for some measurements were observed (eg no volume change in the cerebellum and no cell number change in cortex and DG region), clearly stating where the gene expression data and spine density data come from would help to put these findings in context with the rest of the findings. In general, a discussion on how the different morphological and gene expression changes correlate between regions would be helpful.

3. It would be helpful to add some clarifications on the general health and survival of Tspan7 rats since it may have an effect on early embryonic survival, at least in males (Figure 2G). Is the total litter size also reduced? The weight data is from adults, but are the developmental weight and milestones similar? For the behavioral results, it would be helpful to provide the total investigation time for Figure 3G and I and not just the percentage, to see if the total investigation time is also changed. It would also be useful if some general movement data, for example from an open field test, could be added to the manuscript to control for any changes in the general activity of Tspan7 knockout mice. Information on how the animals were kept (group housed vs single housed) is also important to interpret the behavioral data.

4. For the experiments in Figure 9, there are no controls for each treatment. Treating Tspan7 knockout neurons with the SRC inhibitor would give an indication if this pathway is uncoupled in Tspan7-lacking neurons, and treatment of wildtype neurons with the SRC activator would both confirm that the treatment has the intended effect, and show if SRC activation has different molecular effects in wildtype and knockout neurons. At minimum, treatment with the SRC activator should be performed in wildtype neurons to better evaluate the significance of this result. The method section states that these results were obtained from primary neurons. Is this from cortical neurons, hippocampal neurons, or a mix?

Minor comments

1. It would be helpful if the rationale for choosing the datasets for the expression analysis in figure 1c-f were explained. For example, it seems like the ASD expression levels come from a more than 10-year-old study using 19 ASD cases and 17

controls. Why were not deposited data from recent studies with a much larger sample size used instead? This would make the data more relevant.

2. In Figure 2E-F the figure text does not seem to match the figures. E is genotyping of pups, but just numbers and no genotyping annotation are provided. The figure text states that E and F, n=5, but E shows 8 numbers, and F shows 3 animals per genotype, with no quantification. Please correct this.

3. The text for figures 4f-j and 8d-h says n=5, but there are 6 dots for the Tspan7 knockouts. Which is correct?

5. How were the images in Figure 7 and figure 9 taken and how was the overlap quantified? Especially for some of the zoomed-in examples, it is hard to see how the overlap could have been reliably quantified and a more detailed explanation of the quantification methods would be helpful to evaluate these results.

6. All western blots lack molecular markers; these should be clearly stated. Providing the full uncropped blots would be helpful in evaluating the western blot results. Furthermore, there is no separate loading controls provided for the membrane and cytoplasmic fractions in 8b.

7. The discussion is generally well-written and balanced. However, the author's claim that synaptic plasticity is impaired in primary neurons is not supported by the data. A change in the expression of some synaptic proteins and SRC signaling does not automatically affect synaptic plasticity.

8. Finally, I encourage the authors to be more mindful of language preference within the autism community and avoid the use of terms that are considered harmful, such as disorder, deficit, and disease. The authors should also avoid describing rats as autistic or having ASD as this is a human-only condition (for example: "TSPAN7 knockdown causes ASD in rats" in the synopsis). See for example the recent publication by Bottema-Beutel et al., 2021 (doi/10.1089/aut.2020.0014) on suggestions for language choice in autism research.

Referee Cross-Comments

I agree with the points raised by reviewer 2

Reviewer #2 (Comments to the Authors (Required)):

In this study, Pang et al investigate the role of the tetraspanin transmembrane protein Tspan7 in neurodevelopmental disorders such as autism. Mutations in the human gene coding for Tspan7 are associated with autism and intellectual disability. The authors develop a knock out rat model by Crispr/CAS technology. The Tspan7 gene is located on the X-chromosome. A previous study has analysed Tspan7 ko mice, particularly with a focus on -/y male mice compared to +/y mice. Here, for the rats, the authors analyse -/- females in comparison to +/- female control animals.

The authors convincingly show that the ko rats exhibit behavioural abnormalities, including increased self-grooming, altered social behaviour to strangers, decreased learning abilities e.g. in the Morris water maze. On the level of brain morphology, the total volume of brain areas is reduced; staining for neurons in hippocampus and cortex shows reduced and somewhat scattered staining. In addition, neuronal branching is severely reduced, and the number of dendritic spines is reduced (as shown by Golgi-Cox staining).

As Tspan7 has been shown before to be in a molecular complex with Integrin- β 1, the authors investigate the effect of Tspan7 deficiency on Integrin- β 1 abundance and signalling. Abundance of Integrin- β 1 is reduced, and the phosphorylation of downstream signalling kinases Src and FAK is also strongly reduced in the brains of ko rats. They argue that some of the molecular features of these rats, in particular the reduced expression levels of synaptic proteins PSD-95 and GluR1/2 may be rescued by treating cultured neurons from ko rats with a Src kinase activator.

This is a rather interesting paper characterizing the effects of Tspan7 deficiency in rats. Most of the experiments dealing with the general characterization of the mutant rat line appear to be well done and show significant alterations in brain and neuronal morphology. The relevance for the Integrin- β 1 signalling pathway is also well described.

I have several issues both with the results and with the methodology, as listed below. These points should be clarified.

1. It is unclear throughout the text how samples were obtained for Western blotting. What is "lysis buffer", and how was lysed material separated from insoluble material. As postsynaptic proteins reside in rather insoluble protein complexes (i.e. the postsynaptic density, PSD), this is an important point.

2. I am not at all convinced by the immunocytochemistry pictures in Fig. 7. The labelling appears rather diffuse, and I cannot see the typical synaptic "puncta" one would expect for either PSD-95 or Synaptophysin.

3. Fig. 8: It is unclear how Integrin- β 1 can be in a cytosolic fraction; this is a membrane protein, so we would expect it to be entirely in the membrane fraction. Probably it would make more sense to analyse this in a total lysate fraction, as described e.g. for the AMPA receptor subunits.

4. Fig. 8C: it is completely unclear how the labelling for Integrin in primary neurons was quantified. Is this labelling with an

antibody on intact cells (cell surface labelling)? Or permeabilized cells? How was quantitation performed?

5. Figure 8,9: The authors describe AMPAR1 and AMPAR2 levels; these are not the correct protein names; either GluR1/GluR2 or GluA1/GluA2 should be used.

6. Fig. 9. The authors treat cells with a highly hydrophilic, water soluble peptide to activate Src. It is completely unclear how this may enter cells to activate Src kinase which is localized in the cytosol of neurons. Also the Western Blot data shown in the Figure do not really suggest that the kinase was successfully activated by this treatment.

7. In several instances throughout the text (e.g. page 13), the authors suggest that they analyse "synaptic plasticity". This is not correct, as no experiments involving synaptic plasticity were described in this manuscript.

Response to Reviewers

First of all, we would like to thank the reviewers for the positive, constructive, detailed and valuable comments and suggestions that are very helpful for us.

Reviewer 1:

Major comment 1: Some blots and immunostainings lack statistical support for the claims made in the text. The text states that Tspan7 expression is the highest in the rat brain, but there is no quantification for the supporting data in Figure 1G-H. As the blot itself is unconvincing with no clear Tspan7 band and uneven GAPDH loading for most tissues, providing quantification is essential to support this claim. Similarly, no quantification is provided for the stated altered morphology of neurons shown in Figure 4D. Furthermore, some of the quantifications shown do not support the conclusion drawn in the text. For example, the authors write that Tspan7 expression increases from 0-2 months and then decreases from 2-4 months of age. However, the data in Figure 2b-c does not support this claim: no change is statistically significant and the spread is too large to draw any clear conclusions. On page 7, the authors write that the brain coefficient is slightly decreased compared to WT rats, but the data in Figure 4c show an equal distribution, not at all supporting this claim. The authors should correct these statements.

Reply and revised:

We thank the reviewer very much for pointing out these issues.

Firstly, we have redone the immunoblotting, replaced the images, and added related statistical analysis of Tspan7 protein expression in different tissues (Figure 1I and Figure 1J). In addition, we have also added the statistical analysis of Tspan7 mRNA expression in different tissues (Figure 1H). The modified text is as follows: "...Finally, the TSPAN7 expression in different tissues was confirmed by RT-PCR and western blot, and the results showed that TSPAN7 expression was the highest in rat brain (Fig 1G-J, n=3, * $P<0.05$, ** $P<0.01$ and *** $P<0.001$ vs brain) ...".

For Figure 4D, we have added the quantification of HE images (Fig 4E-I) and also revised the statement as shown below: “The H&E staining showed the gross structure of hippocampus and decreased pyramidal neuron numbers in CA1, CA2, CA3 regions of *Tspan7*^{-/-} rat brains and WT rat brains (Fig 4D-I, n=6 in the WT group and n=5 in the *Tspan7*^{-/-} group, **P*<0.05, ***P*<0.01).”.

For Figure 2B&C, we admit that the spread is indeed large, so we examined more rats for statistical analysis. According to the suggestion of reviewer, we have added the number of rats from n=3 to n= 6 for statistical analysis and revised the conclusion statement for Figure 2B&C in the revised manuscript as shown below: “The western blot indicated the high-level expression of TSPAN7 proteins in rat brain from 0.5 month to 4 months of age, and the relative lower expression of TSPAN7 at birth (Fig 2B-C, n=6, **P*<0.01).”

Finally, we have corrected the inappropriate statements about brain coefficient on page 8. The modified text is as follows: “**The gross anatomy examination showed no significant change in brain coefficient (brain weight/body weight) of *Tspan7*^{-/-} rats compared to WT rats...**”.

Major comment 2: More clarity on which brain regions the data is generated from would be helpful for several figures. Is the data in Figure 5 from the whole brain? If so, why was this chosen instead of for example hippocampus or cortex where the volume and histological differences were identified? I have the same question for the data from Figure 6. What brain regions are these neurons from, and were the brain regions matched closely between wildtype and knockout rats? Since regional effects for some measurements were observed (eg no volume change in the cerebellum and no cell number change in cortex and DG region), clearly stating where the gene expression data and spine density data come from would help to put these findings in context with the rest of the findings. In general, a discussion on how the different morphological and gene expression changes correlate between regions would be helpful.

Revised:

Thank the reviewer very much for the valuable suggestions. We have highlighted the brain regions information throughout the whole manuscript.

1. For figure 1 and figure 2, we used whole brain tissue to detect the expression of **TSPAN7**. We have also added the modified text to the figure legends section as followed: “The protein expression level of TSPAN7 in the **whole brain tissues** from WT rats at age of 0 day, 0.5 month, 1 month, 2 months and 4 months” (on page 29).
2. As showed in Figure 4A&B, we found that hippocampal and cortical volume in *Tspan7*^{-/-} brain were significantly decreased compared with that of WT rats, so we

selected the mixture of the cortex region and hippocampus for subsequent experimental studies including the data of Figure 5. We have also added the modified text to the methods and results section as followed: “The fresh **hippocampal and cortical tissues** were sampled from wild type female Wistar rats (WT) and female *Tspan7* knockout rats (*Tspan7^{-/-}*)...” (on page 15) and “To further explore the molecular events involved in TSPAN7, we analyzed the expression of 9 ASD related genes by RT-qPCR **in the hippocampus and cortex of the rats...**” (on page 8).

3. The data in Figure 6 is also from the mixture of the cortex and hippocampus. We added the modified text to the results and methods section as followed: “To obtain a complete understanding of the changes in neuronal morphology and dendritic spines in *Tspan7^{-/-}* rats, we performed Golgi-Cox staining on the **hippocampus and cortex** of *Tspan7^{-/-}* rats as well as WT rats...” (on page 8) and “The fresh hippocampal and cortical tissues were rinsed with double distilled water...” (on page 18).

4. Yes, the primary neurons are from the cortex and hippocampus and the brain regions were matched closely between wildtype and knockout rats. For figure 8 and 9, we added the brain region information to the results and methods section as followed: “We firstly observed the decreased levels of Integrin β 1 by immunofluorescence staining in **primary *Tspan7^{-/-}* neurons of hippocampus and cortex...**” (on page 9) and “Briefly, the embryonic **hippocampal and cortical tissues** at embryonic day 17...” (on page 19).

5. According to the suggestion of reviewer, we add a discussion on how the different morphological and gene expression changes correlate between regions as followed:

“Previous studies reported that total brain weight and cortical volume were significantly reduced in *Ube3a* and *NLGN3* deficient ASD mice (Varghese et al 2017). like that, we found that the volume of the cortex and hippocampus of the *Tspan7^{-/-}* rats was significantly reduced (Fig 4A-B). The expression of *Ube3a*, *PSD95* and *Nlgn3*, *Syn* were significantly downregulated in *Tspan7^{-/-}* hippocampus and cortex compared with that of WT rats (Fig 5A-D, n=6, * $P < 0.05$), which are important for synapse formation and remodeling (Chen et al 2020, Kim et al 2008, Liu et al 2018, Xu et al 2018). These results suggested that the down-regulation of

synapse-associated genes such as *Ube3a*, *PSD95* and *Nlgn3*, *Syn* in the hippocampus and cortex might be associated with the decrease in hippocampal and cortical volume caused by TSPAN7 knockout, and TSPAN7 deficiency could be involved in pathogenesis of ASD through a synaptic mechanism.”

Major comment 3: It would be helpful to add some clarifications on the general health and survival of Tspan7 rats since it may have an effect on early embryonic survival, at least in males (Figure 2G). Is the total litter size also reduced? The weight data is from adults, but are the developmental weight and milestones similar? For the behavioral results, it would be helpful to provide the total investigation time for Figure 3G and I and not just the percentage, to see if the total investigation time is also changed. It would also be useful if some general movement data, for example from an open field test, could be added to the manuscript to control for any changes in the general activity of Tspan7 knockout mice. Information on how the animals were kept (group housed vs single housed) is also important to interpret the behavioral data.

Revised: Thanks for the reviewer's suggestion.

1. We observed the general health and survival of Tspan7^{-/-} rats at 2-8 weeks of age and found that there was no difference in body weight between Tspan7^{-/-} rats and wild rats from 2 to 8 weeks (Fig S1G). And TSPAN7 rats are now up to 16 months old and their survival is not significantly different from that of WT rats. In addition, we observed the **numbers of littermate** of five pregnant Tspan7^{-/-} rats and found that the average number of pups per pregnant rat was 7.6, which was lower than that of wild-type Wistar rats (They have litter size 10-12, (Figueiró-Filho et al 2014)). Because TSPAN7 is located on the X chromosome, so we crossed Tspan7^{+/-} rat with Tspan7^{-y} rats to obtain **homozygous ko rats**, and we found that the male: female ratio of **homozygous ko rats (4:11)** was significantly lower than that of WT rats (1:1) among all the rats we obtained (Fig 2I). Further experiments should be performed to investigate the effect of Tspan7 knockout on reproductive function and decreased number of **male homozygous rats in the future study**.

2. According to the suggestion of reviewer, we added the total investigation time and general movement data in open field test, Y maze test, social interaction tests and Morris water maze test.

2.1 For grooming behavior, we have added the total moved distance to the figure 3D. The results showed that *Tspan7^{-/-}* rats had no significant changes in the movement compared with WT rats. The modified text in the manuscript expressed as follows: “Firstly, the *Tspan7^{-/-}* rats displayed abnormal self-grooming behavior indicated with increase of self-grooming episodes and times and no significant change in the total distance (Fig 3A-D, n=13, * $P < 0.05$, *** $P < 0.001$, ns).” (on page 7).

2.2 For social interaction tests, we have added the total investigation time and total moved distance to the figure 3I-J. The results showed that the total investigation time and movement of *Tspan7^{-/-}* rats did not change significantly compared with WT rats. The modified text in the manuscript expressed as follows: “However, there was no significantly difference in the total investigation time and distance between the *Tspan7^{-/-}* rats and WT rats (Fig 3I-J, n=13, ns)” (on page 7).

2.3 For Y maze test, we calculated the movement of the Y maze test and the found that there was no significant difference in the movement between *Tspan7^{-/-}* rats and WT rats (Fig 3M). The modified text in the manuscript expressed as follows: “*Tspan7^{-/-}* rats reduced working memory indicated with the decrease of spontaneous alternations in Y-maze test (Fig 3K-L, $n=13$, $**P<0.01$) and did not change the total distance (Fig 3M, $n=13$, ns)” (on page 7).

2.4 For Morris water maze test, we have added the total moved distance to the figure 3Q and found that there was no significant difference in the movement between *Tspan7^{-/-}* rats and WT rats. The modified text in the manuscript expressed as follows: “While there is no significant difference in the total movement between the *Tspan7^{-/-}* rats and WT rats (Fig 3Q, $n=13$, ns).” (on page 7).

3. The rats were housed in groups. Our experiments were approved by IACUC and the rats were housed in groups according to the IACUC requirements. Meanwhile rats and mice are social animals and individual housing constitutes a stressful situation (Manouze et al 2019). Group housing is important for the welfare of social animals and is mandated by law in many countries (Weber et al 2017). In this experiment, we explored behavioral changes such as stereotypical behavior and social deficits in *Tspan7^{-/-}* rats. Because individual housing produces anxiety/depression-like behavioral and cognitive deficits (Bianchi et al 2006), to exclude interference, the animals were housed in groups. The information was added in the methods sections. The modified text in the manuscript expressed as follows: “All rats used in this study were housed in groups and bred in an AAALAC.....” (on page 15).

Major comment 4: For the experiments in Figure 9, there are no controls for each treatment. Treating *Tspan7* knockout neurons with the SRC inhibitor would give an indication if this pathway is uncoupled in *Tspan7*-lacking neurons, and treatment of wildtype neurons with the SRC activator would both confirm that the treatment has the intended effect, and show if SRC activation has different molecular effects in wildtype and knockout neurons. At minimum, treatment with the SRC activator should be performed in wildtype neurons to better evaluate the significance of this result. The method section states that these results were obtained from primary neurons. Is this from cortical neurons, hippocampal neurons, or a mix?

Revised: We agree with this point. According to the reviewer suggestion, we stimulated WT neurons with SRC activator and found that hyperactivation of SRC promoted the expression of downstream GluR1/2, PSD95, SYN, and improved the synaptic integrity in WT neurons. We have added the corresponding results to the figure 9. The modified text in the manuscript expressed as follows: “...Like the inhibitor, in *Tspan7^{-/-}* neurons, *Tspan7* knockout also caused the inhibition of SRC signal and the downregulation of the synaptic proteins and receptors. But when the SRC activator was administrated to the cells, the SRC phosphorylation, as well as the

expression of PSD95, SYN, GluR1 and GluR2, was significantly enhanced or restored in the WT and *Tspan7*^{-/-} neurons ...”. (on page 10).

The primary neurons were isolated from a mix of cortex and hippocampus. We have added the modified text in the methods section as followed: “Briefly, the embryonic hippocampal and cortical tissues at embryonic day 17 (E17) from WT and *Tspan7*^{-/-} rats...” (on page 19).

Minor comment 1: It would be helpful if the rationale for choosing the datasets for the expression analysis in figure 1c-f were explained. For example, it seems like the ASD expression levels come from a more than 10-year-old study using 19 ASD cases and 17 controls. Why were not deposited data from recent studies with a much larger sample size used instead? This would make the data more relevant.

Revised: Thanks for the reviewer's suggestion. We have added the rationale of filtering datasets in the methods. The modified text as followed:

“... “Homo sapiens” AND “tissue” AND “disease” were firstly searched in the GEO datasets and then the data with small sample sizes or no control group were filtered out. Finally, the data was finally identified for analysis, including autism spectrum disorder (ASD, GSE30573), Huntington disease (HD, GSE129473), Alzheimer's disease (AD, GSE36980), Parkinson's disease (PD, GSE20295) ...”

For ASD, we searched the GEO database for sequencing results of brain tissue samples from ASD patients and found a total of 53 items, most of which were blood samples and iPSC sequencing results, while only 5 sets of data were from brain tissue samples of ASD patients, including GSE98581, GSE117776, GSE59288, GSE102741 and GSE30573. Among that, GSE98581, GSE117776 and GSE59288 were either small samples ($n < 3$) or no control group, GSE102741 was the sequencing result of the dorsolateral prefrontal cortex, and no significant difference in TSPAN7 expression was found in the data. A decrease in TSPAN7 expression was found in the temporal cortex detected in the GSE30573 database.

- Large 22q13.3 deletions perturb peripheral transcriptomic and metabolomic profiles in Phelan-McDermid syndrome**
 (Submitter supplied) Peripheral blood transcriptomic data were generated across 68 PMS participants, including Class I sequence variants and mutations (n=33) and Class II mutations (n=35), as well as an age and sex matched control group (n=24), which largely consisted of unaffected siblings (~91%).
 Organism: **Homo sapiens**
 Type: **Expression profiling by high throughput sequencing**
 Platform: GPL16791 92 Samples
 Download data: TXT
 Series: Accession: GSE212096 ID: 200212096
 Published Full text in PMC Similar studies
- Whole-transcriptome analysis of serum L1CAM-captured extracellular vesicles related to autism spectrum disorder**
 (Submitter supplied) L1CAM-captured extracellular vesicles (LCEVs) were isolated and characterized meticulously. Whole-transcriptome of LCEVs was analyzed by lncRNA microarray and RNA-Sequencing. RNAs expressed differently in LCEVs from ASD sera vs. TD sera were screened, analyzed, and further validated.
 Organism: **Homo sapiens**

The search details box shows the search query: ("autistic disorder"[Mesh Terms] OR AUTISM[All Fields]) AND "Homo sapiens"[porgn] AND ("gse"[Filter] AND ("Expression profiling by high throughput sequencing"[Filter] OR

For **Huntington disease**, we have updated the Huntington's data with a total of 85 samples tested in the 2019 GSE129473 data, and we found that the TSPAN7 expression in the prefrontal cortex of HD patients was decreased and updated the modified figure to Figure 1D.

For **AD**, we used the same screening method for sequencing results of brain tissue from AD patients. Among the screened data, GSE36980 was selected because of its large sample size and comprehensive intracerebral partitioning, with a total of 80 cases tested, three brain regions frontal cortex, temporal cortex, and hippocampus samples.

For **PD**, we used the same screening method for sequencing results of brain tissue from PD patients. GSE20295 included 93 samples containing multiple brain regions. So we selected the dataset for further analysis.

Minor comment 2: In Figure 2E-F the figure text does not seem to match the figures. E is genotyping of pups, but just numbers and no genotyping annotation are provided. The figure text states that E and F, n=5, but E shows 8 numbers, and F shows 3 animals per genotype, with no quantification. Please correct this.

Revised: We thank the reviewer for pointing out this mistake. We have corrected this. In Figure 2E legend, the modified text in the manuscript as followed: "**N was negative indicated WT. Number 4, 7, 8 are WT, number 2, 3 are heterozygotes, number 1, 5, 6**

are homozygotes (n=8)". In Figure 2, we have added the quantification of Figure 2F to the figure and legend. The modified section as followed: "F-G The deletion of TSPAN7 protein in brain tissues of *Tspan7^{-/-}* rats was detected by western blot with the anti-TSPAN7 and quantified with a software of Image J (n=3)."

Minor comment3: The text for figures 4f-j and 8d-h says n=5, but there are 6 dots for the *Tspan7* knockouts. Which is correct?

Revised: We thank the reviewer for pointing out this issue. In figures 4f-j (Now change it to figures 4k-o), we performed a quantitative analysis of Nissl staining on 5 rats in the WT group and 6 rats in the *Tspan7^{-/-}* group. In figures 8d-h, there are 6 rats for both WT group and *Tspan7^{-/-}* group were analyzed. We have modified the text for figures. The text in the manuscript as followed: "Figure 4 F-J The Nissl positive cells (blue) were counted in the cortex and hippocampus of CA1, CA2, CA3 and DG region, n=5 for WT group and n=6 for *Tspan7^{-/-}* group, Scale bar = 20 μ m" and "In Figure 8 D-L The expression levels of the Integrin β 1 in membrane and cytosol fractions, the phosphorylated FAK, SRC, PSD95, SYN, GluR1/2 in total lysates were quantified with a software of Image J. n=6 each group."

Minor comment 4: How were the images in Figure 7 and figure 9 taken and how was the overlap quantified? Especially for some of the zoomed-in examples, it is hard to see how the overlap could have been reliably quantified and a more detailed explanation of the quantification methods would be helpful to evaluate these results.

Revised: We thank the reviewer for the valuable suggestions. We replaced the images with the higher quality images. and supplemented the description about the quantitative details, and added the modified text in the method section. The text in revised the manuscript as followed:

"Quantification of post-synaptic and presynaptic puncta marked by immunostaining. The post-synaptic and presynaptic puncta were analyzed as previously (Block et al 2022, Ippolito & Eroglu 2010). For each area of brain, using both the 488-nm channel to detect SYN and the 555-nm channel to detect the PSD95. A z-stack for each brain

section was collected in LAS X software for a total depth of 5 μm (15 \times 0.33 μm optical sections) at 63 \times magnification using a Multiphoton Laser Scanning Microscopy (Leica). Maximum image projections (MIPs) were generated for groups of 3 consecutive optical sections yielding 5 MIPs/section each representing 1 μm of depth. Maximum projections of 3 consecutive optical sections were generated using ImageJ. The Puncta Analyzer Plugin (available at: <https://doi.org/10.5281/zenodo.6800214>) for ImageJ was used to count the number of colocalized synaptic puncta and the number of synaptic puncta produced by Puncta Analyzer.” (on page 20).

The modified figure 7 and 9 as followed:

Figure7

Figure9

Minor comment 5: All western blots lack molecular markers; these should be clearly stated. Providing the full uncropped blots would be helpful in evaluating the western blot results. Furthermore, there is no separate loading controls provided for the membrane and cytoplasmic fractions in 8b.

Revised: According to the reviewer suggestion. We added the molecular markers and

provided the full cropped blots as followed.

Figure 9A

Then we used caveolin-1 as loading controls for membrane proteins and added this result to Figure 8b. The results showed that a significant reduction of Integrin $\beta 1$ in total lysates and plasma membrane fraction in the brain tissues of *Tspan7*^{-/-} rats was then confirmed by western blot, while the levels of Integrin $\beta 1$ in cytosol fraction was not changed (Fig 8B&D&E&F, n=6, ** $P < 0.01$, n.s).

Minor comment 6: The discussion is generally well-written and balanced. However, the author's claim that synaptic plasticity is impaired in primary neurons is not supported by the data. A change in the expression of some synaptic proteins and SRC signaling does not automatically affect synaptic plasticity.

Revised: We agree with this reviewer that this was not properly explained. We have modified the “synaptic plasticity” to “synaptic integrity”. The modified text as followed:

“...Then, we found TSPAN7 deletion interrupted the Integrin $\beta 1$ /FAK/SRC signal pathway that was followed by the downregulation of PSD95, SYN and GluR1/2, which are key synaptic integrity-related proteins. Furthermore, re-activation of SRC restored the expression of synaptic integrity-related proteins in primary neurons of TSPAN7 knockout brains...” (on page 2).

Minor comment 7: Finally, I encourage the authors to be more mindful of language preference within the autism community and avoid the use of terms that are considered harmful, such as disorder, deficit, and disease. The authors should also avoid describing rats as autistic or having ASD as this is a human-only condition (for example: "TSPAN7 knockdown causes ASD in rats" in the synopsis). See for example the recent publication by Bottema-Beutel et al., 2021 (doi/10.1089/aut.2020.0014) on suggestions for language choice in autism research.

Revised: We thank the reviewer for pointing out this issue. We modified the language according to the paper (doi/10.1089/aut.2020.0014) and cited on page 4. They are listed as follows:

1. "TSPAN7 knockdown causes **ASD-like behaviors** in rats" (on page 3).
2. "...most of the variation **that increase likelihood of ASD** is believed to have its origin in a complex interaction between genetic and environmental factors..." (on page 4).
3. "A variety of environmental factors had been found to increase the **likelihood of ASD**" (on page 4).
4. "The main **specific autistic characteristics** are social communication deficits, interest deficits, repetitive stereotyped behavior and mental retardation in some cases" (on page 4).
5. "Consistent with *Tspan7^{-y}* mice, the *Tspan7^{-/-}* rats showed both of **ASD-like** and **ID-like** phenotypes with decreased sociability..." (on page 11).
6. "our finding indicated that TSPAN7 deficiency caused both of **ASD-like** and **ID-like** phenotypes..." (on page 13).
7. "Taken together, our results suggested TSPAN7 knockout caused **ASD-like** and **ID-like behaviors** in rats..." (on page 2).

Reviewer #2

Comment 1: It is unclear throughout the text how samples were obtained for Western blotting. What is "lysis buffer", and how was lysed material separated from insoluble material. As postsynaptic proteins reside in rather insoluble protein complexes (i.e. the postsynaptic density, PSD), this is an important point.

Revised: We thank the reviewer for these suggestions and apologized for the inadequate description in the methods. The "lysis buffer" is lysis buffer **containing RIPA buffer (50mM Tris, 150mM NaCl, 1% Triton X-100, 1% sodium deoxycholate, 0.1% SDS, P0013B, Beyotime)**, a protease inhibitor mixture (87785, Thermo Scientific), a phosphatase inhibitor cocktail (78420, Thermo Scientific), and 1mM phenylmethyl-sulfonylfluoride (PMSF, 36978, Thermo Scientific).

We agree with the reviewer that postsynaptic proteins reside in rather insoluble protein complexes. But as reported by Fiszer and de Robertis, the detergent Triton X-100 could solubilize most of the membrane components of the synaptosomes in 1967 (Fiszer & Robertis 1967). And then in 1977, Cohen improved the method for the isolation of a purified PSD fraction from brain by the use of Triton X-100, as well as observations on its structure and its chemical constituents (Cohen et al 1977).

Theoretically, therefore, RIPA buffer containing 1% Triton X-100 that we used in the present study could dissolve the PSD95 proteins. Meanwhile, in various previous studies RIPA lysis buffer containing 1% Triton X-100 was used to obtain brain tissue lysate and subsequent western blot was performed to detect PSD95 proteins (Kalia & Salter 2003, Murru et al 2021).

Comment 2: I am not at all convinced by the immunocytochemistry pictures in Fig. 7. The labelling appears rather diffuse, and I cannot see the typical synaptic "puncta" one would expect for either PSD-95 or Synaptophysin.

Revised: We thank the reviewer for these suggestions and apologized for the low-quality pictures of the figure 7 and inadequate description of the method. We replaced the images Fig. 7 with the higher quality images, supplemented the quantitative details, and added the modified text in the method section. The text in the

manuscript as followed:

“Quantification of post-synaptic and presynaptic puncta marked by immunostaining. The post-synaptic and presynaptic puncta were analyzed as previously (Block et al 2022, Ippolito & Eroglu 2010). For each area of brain, using both the 488-nm channel to detect SYN and the 555-nm channel to detect the PSD95. A z-stack for each brain section was collected in LAS X software for a total depth of 5 μm (15 \times 0.33 μm optical sections) at 63 \times magnification using a Multiphoton Laser Scanning Microscopy (Leica). Maximum image projections (MIPs) were generated for groups of 3 consecutive optical sections yielding 5 MIPs/section each representing 1 μm of depth. Maximum projections of 3 consecutive optical sections were generated using ImageJ. The Puncta Analyzer Plugin (available at: <https://doi.org/10.5281/zenodo.6800214>) for ImageJ was used to count the number of colocalized synaptic puncta and the number of synaptic puncta produced by Puncta Analyzer.” (on page 20).

The modified figure 7 as followed:

Comment 3: Fig. 8: It is unclear how Integrin- β 1 can be in a cytosolic fraction; this is a membrane protein, so we would expect it to be entirely in the membrane fraction. Probably it would make more sense to analyse this in a total lysate fraction, as described e.g. for the AMPA receptor subunits.

Revised:

We agree with the reviewer. But we analyzed the expression of Integrin- β 1 in the cytoplasm fraction, membrane fraction and total cell lysate. We found that just small amount of Integrin- β 1 was present in cytoplasm fraction, and most Integrin- β 1 proteins appeared in membrane fraction, suggesting that Integrin- β 1 is a membrane protein.

We further found that Integrin- β 1 proteins from the total cell lysate and from the membrane fraction of *Tspan7*^{-/-} rats were both significantly decreased when compared with that of WT rats, whereas the levels of Integrin β 1 in cytosol fraction was not changed. The results indicated that Integrin- β 1 on cell membrane is involved in TSPAN7-regulated downstream signaling pathways. The modified Figure 8 and text in the manuscript as followed: “A significant reduction of Integrin β 1 in total lysates and plasma membrane fraction in the brain tissues of *Tspan7*^{-/-} rats was then confirmed by western blot, while the levels of Integrin β 1 in cytosol fraction was not changed (Fig 8B&D&E&F, n=6, ** P <0.01, n.s).”(on page 9).

In addition, the data from the database of Genecards and The Human Protein Atlas showed that Integrin- β 1 is mainly localized to the cell membrane, but Endoplasmic reticulum also contains a small amount of Integrin- β 1, which could explain why we detected small amount of Integrin- β 1 proteins in the cytoplasm.

Comment 4: Fig. 8C: it is completely unclear how the labelling for Integrin in primary neurons was quantified. Is this labelling with an antibody on intact cells (cell surface labelling)? Or permeabilized cells? How was quantitation performed?

Revised: We thank the reviewer for pointing out this issue and apologized for the inadequate description of the method.

The Integrin-β1 antibody was labelled on Triton X-100-permeabilized cells. And we have added the permeabilizer information and quantitation method to the methods section in the revised manuscript as followed “...Briefly, the primary neurons were fixed in paraformaldehyde (4%) for 20 mins at room temperature and penetrated with 0.1% Triton X-100 for 5min. Then the slices washed 3 times with PBS for 5min... The images were quantified using ImageJ software. Images were first converted to 8-bit gray scale and binary threshold to highlight a positive staining. Finally, Percentage of area positive for Integrin-β1 were measured by the image j” (on page

17-18). We have modified the manuscript and highlight these in red.

Comment 5: Figure 8,9: The authors describe AMPAR1 and AMPAR2 levels; these are not the correct protein names; either GluR1/GluR2 or GluA1/GluA2 should be used.

Revised: We thank the reviewer for the suggestion. We modified “AMPAR1 and AMPAR2” to “GluR1/GluR2” in the manuscript and figures and highlight these in red.

Comment 6: Fig. 9. The authors treat cells with a highly hydrophilic, water soluble peptide to activate Src. It is completely unclear how this may enter cells to activate Src kinase which is localized in the cytosol of neurons. Also the Western Blot data shown in the Figure do not really suggest that the kinase was successfully activated by this treatment.

Revised:

We thank the reviewer for pointing out this issue. We have added the experiments and revised the description of activator administration method in the methods section in the revised manuscript.

1. According to the suggestion of reviewer 1, we have used SRC activators to stimulate WT primary neurons and found that that SRC activator significantly increased SRC phosphorylation and increase the expression of downstream synapse-associated proteins (Figure 9A&B).

According to the method of Src activator administration to cells previously reported (Park et al 2015), in our study, the cells were starved for 2 hours before by the treatment with serum or a 10 μ M Src activator (EPQpYEEIPIYL, HY-P3279, MedChemExpress). Thus, we added the modified text to the methods section as followed: “...As previously reported (Park et al 2015), the WT *Tspan7*^{-/-} neurons were starved using DMEM-F-12 medium supplemented with 0.1% BSA for 2 hours followed by stimulation with serum or a 10 μ M Src activator (EPQpYEEIPIYL, HY-P3279, MedChemExpress)) for 1 hour ...”.

2. Explanation : As previously reported, the off-domain positive charge on the guanidine group of arginine is thought to interact extensively with negatively charged cell membranes, and arginine-rich peptides such as TAT peptides (GRKKRRQRRRRPSQ) are eventually phagocytosed by macrophages mainly through binding to glycosaminoglycans (GAGs) on the cell surface, such as heparan sulfate, or by adsorption to the glycerol backbone region of lipid bilayers (Yang & Hinner 2015). Besides specialized proteins and peptides mentioned above, endocytosis mechanisms at the level of the lipid bilayer can transport molecules from the outside across the cell membrane, in an energy-dependent manner (Walrant et al 2017).

Therefore, the peptides SRC activator we used in the present study may enter the cell by endocytosis and act as a SRC activator that increased the phosphorylation of SRC and the expression of downstream synapse-associated proteins.

Comment 7: In several instances throughout the text (e.g. page 13), the authors suggest that they analyse "synaptic plasticity". This is not correct, as no experiments involving synaptic plasticity were described in this manuscript.

Revised: We agree with this reviewer that this was not properly explained. We have modified the "synaptic plasticity" to "synaptic integrity". We have made 8 changes throughout the text, on pages 2, 5, 10 and 13. The modified text as followed (e.g. page

13):

“Our results showed that inhibition of SRC phosphorylation by Bosutinib reduced the expression of PSD95, SYN and GluR1/2, and impaired synaptic integrity in primary WT neurons... The Integrin β 1/FAK/SRC signal pathway was interrupted by TSPAN7 deficiency and resulted in the downregulation of PSD95, SYN, GluR1/2, which were essential protein of synaptic integrity...”.

Reference

- Bianchi M, Fone KF, Azmi N, Heidbreder CA, Hagan JJ, Marsden CA. 2006. Isolation rearing induces recognition memory deficits accompanied by cytoskeletal alterations in rat hippocampus. *The European journal of neuroscience*. 24(10):2894-2902. doi:10.1111/j.1460-9568.2006.05170.x
- Block CL, Eroglu O, Mague SD, Smith CJ, Ceasrine AM, Sriworarat C, Blount C, Beben KA, Malacon KE, Ndubuizu N, et al. 2022. Prenatal environmental stressors impair postnatal microglia function and adult behavior in males. *Cell reports*. 40(5):111161. doi:10.1016/j.celrep.2022.111161
- Chen HR, Chen CW, Mandhani N, Short-Miller JC, Smucker MR, Sun YY, Kuan CY. 2020. Monocytic infiltrates contribute to autistic-like behaviors in a two-hit model of neurodevelopmental defects. *The Journal of neuroscience : the official journal of the Society for Neuroscience*. 40(49):9386-9400. doi:10.1523/jneurosci.1171-20.2020
- Cohen RS, Blomberg F, Berzins K, Siekevitz P. 1977. The structure of postsynaptic densities isolated from dog cerebral cortex. I. Overall morphology and protein composition. *The Journal of cell biology*. 74(1):181-203. doi:10.1083/jcb.74.1.181
- Figueiró-Filho EA, Aydos RD, Senefonte FR, Ferreira CM, Pereira EF, Oliveira VM, Menezes GP, Bósio MA. 2014. Effects of enoxaparin and unfractionated heparin in prophylactic and therapeutic doses on the fertility of female wistar rats. *Acta cirurgica brasileira*. 29(7):410-416. doi:10.1590/s0102-86502014000700001

- Fiszer S, Robertis ED. 1967. Action of triton x-100 on ultrastructure and membrane-bound enzymes of isolated nerve endings from rat brain. *Brain research*. 5(1):31-44. doi:10.1016/0006-8993(67)90217-x
- Ippolito DM, Eroglu C. 2010. Quantifying synapses: An immunocytochemistry-based assay to quantify synapse number. *Journal of visualized experiments : JoVE*. (45) doi:10.3791/2270
- Kalia LV, Salter MW. 2003. Interactions between src family protein tyrosine kinases and psd-95. *Neuropharmacology*. 45(6):720-728. doi:10.1016/s0028-3908(03)00313-7
- Kim HG, Kishikawa S, Higgins AW, Seong IS, Donovan DJ, Shen Y, Lally E, Weiss LA, Najm J, Kutsche K, et al. 2008. Disruption of neurexin 1 associated with autism spectrum disorder. *American journal of human genetics*. 82(1):199-207. doi:10.1016/j.ajhg.2007.09.011
- Liu CX, Li CY, Hu CC, Wang Y, Lin J, Jiang YH, Li Q, Xu X. 2018. Crispr/cas9-induced shank3b mutant zebrafish display autism-like behaviors. *Molecular autism*. 9:23. doi:10.1186/s13229-018-0204-x
- Manouze H, Ghestem A, Poillierat V, Bennis M, Ba-M'hamed S, Benoliel JJ, Becker C, Bernard C. 2019. Effects of single cage housing on stress, cognitive, and seizure parameters in the rat and mouse pilocarpine models of epilepsy. *eNeuro*. 6(4) doi:10.1523/eneuro.0179-18.2019
- Murru L, Ponzoni L, Longatti A, Mazzoleni S, Giansante G, Bassani S, Sala M, Passafaro M. 2021. Lateral habenula dysfunctions in tm4sf2(-/y) mice model for neurodevelopmental disorder. *Neurobiology of disease*. 148:105189. doi:10.1016/j.nbd.2020.105189
- Park SY, Yang JS, Schmider AB, Soberman RJ, Hsu VW. 2015. Coordinated regulation of bidirectional copi transport at the golgi by cdc42. *Nature*. 521(7553):529-532. doi:10.1038/nature14457
- Varghese M, Keshav N, Jacot-Descombes S, Warda T, Wicinski B, Dickstein DL, Harony-Nicolas H, De Rubeis S, Drapeau E, Buxbaum JD, et al. 2017. Autism spectrum disorder: Neuropathology and animal models. *Acta neuropathologica*. 134(4):537-566. doi:10.1007/s00401-017-1736-4

- Walrant A, Cardon S, Burlina F, Sagan S. 2017. Membrane crossing and membranotropic activity of cell-penetrating peptides: Dangerous liaisons? *Accounts of chemical research*. 50(12):2968-2975. doi:10.1021/acs.accounts.7b00455
- Weber EM, Dallaire JA, Gaskill BN, Pritchett-Corning KR, Garner JP. 2017. Aggression in group-housed laboratory mice: Why can't we solve the problem? *Lab animal*. 46(4):157-161. doi:10.1038/labam.1219
- Xu X, Li C, Gao X, Xia K, Guo H, Li Y, Hao Z, Zhang L, Gao D, Xu C, et al. 2018. Excessive ube3a dosage impairs retinoic acid signaling and synaptic plasticity in autism spectrum disorders. *Cell research*. 28(1):48-68. doi:10.1038/cr.2017.132
- Yang NJ, Hinner MJ. 2015. Getting across the cell membrane: An overview for small molecules, peptides, and proteins. *Methods in molecular biology* (Clifton, NJ). 1266:29-53. doi:10.1007/978-1-4939-2272-7_3

December 1, 2022

RE: Life Science Alliance Manuscript #LSA-2022-01616-TR

Prof. Lianfeng Zhang
Chinese Academy of Medical Sciences & Peking Union Medical College
Key Laboratory of Human Disease Comparative Medicine, Ministry of Health
#5 Panjiayuan Nanli, Chaoyang District, Beijing, China
beijing 100021
China

Dear Dr. Zhang,

Thank you for submitting your revised manuscript entitled "Integrin β 1/FAK/SRC signal pathway is involved in autism spectrum disorder in Tspan7 knockout rats". We would be happy to publish your paper in Life Science Alliance pending final revisions necessary to meet our formatting guidelines.

- please include the full blots you provided in your point by point rebuttal letter in the supplementary figures, and address the final Reviewer 2's minor comments
- please add ORCID ID for both corresponding authors; you should have received instructions on how to do so
- please add the Twitter handle of your host institute/organization as well as your own or/and one of the authors in our system

Figure Check:

- Please add scale bars to Figure 4D and Figure 4J
- Please add scale bars to Figure 4A, if possible

A. FINAL FILES:

B. MANUSCRIPT ORGANIZATION AND FORMATTING:

Sincerely,

Reviewer #1 (Comments to the Authors (Required)):

I thank the authors for fully addressing my comments, and the additional data and clarifications provided have largely answered all my questions. My only ongoing concern is the western blot data based on the full blots provided. Especially the Integrin beta-1 antibody looks to be highly unspecific, which makes it difficult to trust the data fully. I would suggest including the full blots in the supplementary figures and not only providing them to the reviewer.

I have two additional minor comments:

I would suggest changing the wording of the summary blurb. It now says "TSPAN7 knockout causes ASD in rats through the Integrin β 1/FAK/SRC signal pathway,". Rats cannot have ASD. I would suggest changing it to ASD-like phenotypes as it is written in the text below.

There seem to be a mistake on page 12 and supplementary figure 2, where one of the PKC isoforms is written as PKC ϵ (pound sign). I assume this should be PKC δ ?

Despite these minor comments, I am satisfied with the revisions and have no objections to the acceptance of this manuscript for publication.

Reviewer #2 (Comments to the Authors (Required)):

Most of my comments have been addressed in a satisfying manner. I have no further objections.

Response to Reviewer 1

First of all, we would like to thank the reviewer for his/her patience, carefulness and preciseness. We have uploaded the full blots (including the Integrin beta-1) to the supplementary (Source Data for Figure).

Minor comment 1: I would suggest changing the wording of the summary blurb. It now says "TSPAN7 knockout causes ASD in rats through the Integrin β 1/FAK/SRC signal pathway,". Rats cannot have ASD. I would suggest changing it to ASD-like phenotypes as it is written in the text below.

Reply and revised: We thank the reviewer for the valuable suggestions and changed the text in the summary blurb. The text in revised the manuscript as followed: "TSPAN7 knockout causes ASD-like phenotypes in rats through the Integrin β 1/FAK/SRC signal pathway,".

Minor comment 2: There seem to be a mistake on page 12 and supplementary figure 2, where one of the PKC isoforms is written as PKC \pounds (pound sign). I assume this should be PKC δ ?

Reply and revised: We thank the reviewer very much for pointing out these issues. We have corrected the text and supplementary figure. The revised text and supplementary figure as followed: "...we did not observe any significant changes of phosphorylation on ERK, PKC ϵ (PKC epsilon), PKC α and expression levels of ERK, PKC ϵ , PKC α , PICK1, NMDAR1 in *Tspan7*^{-/-} brain compared to WT rats...".

December 2, 2022

RE: Life Science Alliance Manuscript #LSA-2022-01616-TRR

Prof. Lianfeng Zhang
Chinese Academy of Medical Sciences & Peking Union Medical College
Key Laboratory of Human Disease Comparative Medicine, Ministry of Health
#5 Panjiayuan Nanli, Chaoyang District, Beijing, China
beijing 100021
China

Dear Dr. Zhang,

Thank you for submitting your Research Article entitled "Integrin β 1/FAK/SRC signal pathway is involved in autism spectrum disorder in Tspan7 knockout rats". It is a pleasure to let you know that your manuscript is now accepted for publication in Life Science Alliance. Congratulations on this interesting work.

DISTRIBUTION OF MATERIALS:

Again, congratulations on a very nice paper. I hope you found the review process to be constructive and are pleased with how the manuscript was handled editorially. We look forward to future exciting submissions from your lab.

Sincerely,
